# Vertical Attention: Automatic Exploration of Inter-Layer Connections in Transformer-based Language Models

## Abstract

The Transformer architecture has become the de facto standard across natural language processing and other modalities, demonstrating strong generality and performance. However, the conventional design, which stacks attention and feed-forward blocks sequentially, is not guaranteed to be optimal. More expressive inter-layer connectivity patterns, such as parallelization or skip connections across distant layers, may exist but are difficult to discover through manual exploration. In this work, we propose a method to automatically learn inter-layer network paths during training. Our approach introduces a small number of parameterized attention modules at the beginning of each layer, which are interpreted as inter-layer connections, and optimizes these paths end-to-end. Through large-scale experiments with LLaMA-style models ranging from 50M to 300M parameters pretrained on 20B tokens, we show that our method consistently achieves lower pretraining loss than vanilla Transformers and competitive baselines. Analysis of the learned attention maps reveals intriguing patterns, such as strong interactions from lower to higher layers and attention sparsity in the middle layers. Furthermore, logit lens analysis demonstrates that our Transformer almost entirely postpones output prediction until the final layer, exhibiting fundamentally different internal behavior from that of a vanilla Transformer. Finally, we validate the effectiveness of the proposed architecture in downstream tasks in a few-shot in-context learning environment, confirming its applicability and utility.

## 1 Introduction

Transformer architecture (Vaswani et al., 2017) has rapidly become the backbone of modern deep learning, delivering state-of-the-art performance in natural language processing (NLP) and other modalities(Brown et al., 2020; Dosovitskiy et al., 2021; Baevski et al., 2020; Jaegle et al., 2021). Its widespread success has established it as the de facto standard for large-scale models. Transformer architecture stacks multiple blocks in a sequential manner, and each block is built by a multi-head attention (MHA) module and a Multi-Layer Perceptron (MLP) module.

While this simple design has proven highly effective, there is no theoretical guarantee that such a sequential arrangement is optimal. Alternative inter-block organizations—such as parallel connections of multiple blocks or long-range skip connections—may provide stronger expressiveness or better training stability. For example, some recent works have introduced additional inter-layer interactions, such as Adaptive Layerwise Attention (Verma & Pilanci, 2024) and Skip Layer Attention (Chen et al., 2024). These methods demonstrate the value of exchanging information across distant layers. However, manually searching this design space through heuristic exploration is infeasible.

This study proposes a novel framework that treats inter-layer connections as a parameterized attention mechanism. Specifically, we insert a new module at the beginning of each layer that computes an attention distribution over previous layers with a small number of parameters, thereby enabling the model to dynamically select and weight network paths during training. This perspective allows inter-block connectivity to be optimized end-to-end, jointly with the rest of the model parameters.

We validate our approach through extensive large-scale pretraining experiments on LLaMA3 architectures (Dubey et al., 2024) ranging from 50M to 300M parameters pre-trained on 20B to-

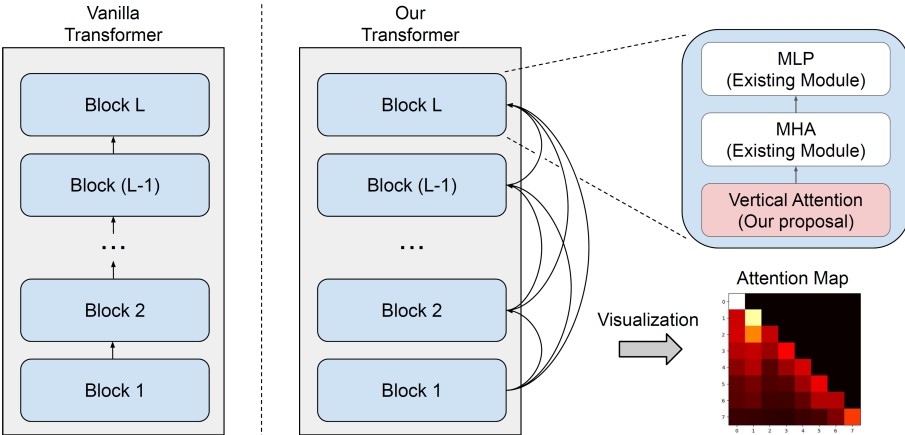

Figure 1: Overview of our proposal. Conventional or vanilla Transformer architecture connects each block sequentially. In contrast, our Transformer flexibly interconnects every block through "Vertical Attention" modules. This attention is automatically optimized during training from data.

kens. Across all scales, our method consistently achieves lower pretraining loss curves compared to both the vanilla Transformer and strong baselines. Analysis of the learned inter-layer attention maps reveals intriguing structural patterns such as strong interactions from lower to higher layers and attention sparsity in the middle layers. Furthermore, logit lens analysis (Wendler et al., 2024) demonstrates that our Transformer almost entirely postpones output prediction until the final layer, exhibiting fundamentally different internal behavior from that of a vanilla Transformer. Finally, we confirm the generality of our method by demonstrating improvements in downstream tasks under few-shot in-context learning (Brown et al., 2020) settings. Our contributions are summarized as follows:

- This study introduces a Vertical Attention mechanism that enables a Transformer to learn a better inter-layer connection architecture by itself during pre-training in an end-to-end manner.

- Experiments demonstrate that our architectur consistently improves in pre-training loss and downstream few-shot in-context learning performance across LLaMA-style models (50M–300M).

- Further analysis reveals meaningful structural patterns in learned attention maps and distinct internal behavior via logit lens analysis, highlighting delayed output prediction until the final layer.

## 2 RELATED WORK

**Transformer Variants**   Since its introduction, the Transformer (Vaswani et al., 2017) has been widely adapted across modalities, including NLP (Brown et al., 2020) vision (Dosovitskiy et al., 2021), speech (Baevski et al., 2020), and multimodal architectures (Jaegle et al., 2021). The recent advancement of LLMs mostly rely on the variants of Transformer architecture, with many types of improvement such as Mixture of Experts (MoE) (Fedus et al., 2022), Rotary Positional Embeddings (Su et al., 2024), and pre-layer normalization (Xiong et al., 2020). Almost all the recent LLMs adopt the canonical sequential connections of blocks.

**Cross-Layer Connections**   Residual connections (He et al., 2016) have been introduced in standard Transformers, but they are limited to a structure that allows connections only to the immediately preceding block. Some recent works have explicitly considered more distant inter-layer interactions.

Skip Layer Attention (Chen et al., 2024) introduces a mechanism in transformer architectures in which every layer directly attends to N-distant previous layers, enabling the model to capture dependencies between high-level abstract features and low-level detailed features effectively. More specifically, at each layer, h out of H heads in the MHA module attend to the features of N-distant previous layers, while the rest of (H-h) heads attend to the previous tokens of current layers with usual attentions.

Table 1: Related work on distant inter-layer connections for Transformer-based language models.

| Method | Description | Flexibility of the inter-layer connections |
|---|---|---|
| Skip Layer Attention (Chen et al., 2024) | Every layer attends to N-distant previous layer. | ✗ No (pre-defined) |
| Adaptive Layerwise Attention (Verma & Pilanci, 2024) | Only the final layer attends to features of every K-th previous intermediate layers. | ✗ No (pre-defined) |
| From Byte to Ideas (Videau et al., 2025) | Deeper layers connect to shallower layers like U-Net architecture. | ✗ No (pre-defined) |
| Vertical Attention (Ours) | Learn inter-layer connections at pre-training. | ✓ Yes |

Adaptive Layerwise Attention (Verma & Pilanci, 2024) introduces a mechanism in transformer-based LLMs where only the final layer dynamically attends to outputs from every K-th previous intermediate layers. Concretely, the final layer computes attention weights for the intermediate hidden representations by the inner dot product between the query of the final layer and the key of the previous layers. These two methods demonstrate the utility of exchanging information across distant layers. However, the methods still pre-define connectivity patterns rather than learning them as part of training, limiting the optimal connectivity search.

**Architecture Search** Automated searching for better network architecture has been a long-standing research area called Network Architecture Search (NAS) (Elsken et al., 2019). For example, AutoFormer (Chen et al., 2021) and TF-TAS (Zhou et al., 2022) employ neural architecture search to discover efficient Transformer variants. While effective, these methods often rely on constrained search spaces or predefined wiring assumptions, e.g., the number of heads in MSAs, the ratio of MSAs or MLPs, limiting their ability to directly optimize inter-layer connectivity.

## 3 METHOD

Our approach introduces a Vertical Attention mechanism into the conventional Transformer block components. As shown in Figure 1, each layer consists of three modules: **Vertical Attention (VA)**, Multi-Head Attention (MHA; In contrast to our method, this could referd to as horizontal attention), and an Multi-layer Perceptron (MLP). We do not modify MHA and MLP modules, which already exist in vanilla Transformer. This section explains the mechanism of Vertical Attention module.

In Vertical Attention, we compute inter-layer attention using a parameterized softmax function without performing inner products between query and key vectors across layers. The hidden representations of multiple layers are aggregated through a weighted average, where the weights are determined by the constant attention scores across samples.

Let $L$ be the number of Transformer layers and $h^i \in \mathbb{R}^d$ be the representation produced at layer $i$ (for $i = 1, \ldots, L$). For each target layer $l$ we introduce a set of learnable scalar scores

$$\mathbf{s}_l = [s_{l,1}, s_{l,2}, \ldots, s_{l,l}]^\top \in \mathbb{R}^l. \tag{1}$$

We convert these scores into normalized attention weights via a softmax function:

$$\alpha_{l,i} = \frac{\exp(s_{l,i})}{\sum_{j=1}^{l} \exp(s_{l,j})} \qquad \text{for } i = 1, \ldots, l. \tag{2}$$

When mixing different representations across layers, it is essential to take the magnitude of the representations into account. Specifically, we reweight the softmax distribution by dividing with the L2 norm of $h^i$. Formally, we define

$$\tilde{\alpha}_{l,i} = \frac{\alpha_{l,i}/\|h^i\|_2}{\sum_{j=1}^{l} \alpha_{l,j}/\|h^j\|_2} \qquad \text{for } i = 1, \ldots, l. \tag{3}$$

Here, $\tilde{\alpha}_{l,i}$ represents the final normalized weight, which incorporates both the learned scalar scores and the relative magnitude of the hidden representations. The vertical-aggregated representation for layer $l$ is the weighted average of lower (and current) layer outputs:

$$v^{(l)} \;=\; \sum_{i=1}^{l} \tilde{\alpha}_{l,i}\, h^i. \tag{4}$$

We feed $v^{(l)}$ into the usual horizontal self-attention and subsequent MLP of layer $l$. Using $\mathrm{MHA}(\cdot)$ for the standard intra-layer (token) self-attention and $\mathrm{MLP}(\cdot)$ for the position-wise feed-forward:

$$\tilde{h}^{(l)} = \mathrm{MHA}\left(v^{(l)}\right) \tag{5}$$

$$h_{\mathrm{out}}^{(l)} = \mathrm{MLP}\left(\tilde{h}^{(l)}\right) \tag{6}$$

This design requires only a small number of additional parameters. Specifically, the number of new parameters for our method is given by:

$$\sum_{l=1}^{L} l \;=\; \frac{L(L+1)}{2}, \tag{7}$$

where L is the number of layers. For example, in the case that $L = 12$, our method introduces only 78 additional parameters in total.

## 4 EXPERIMENT

### 4.1 SETTING

**Model Architecture**   We adopt a model architecture of Llama 3 (Dubey et al., 2024), one of the recent state-of-the-art decoder-only Transformer model with several architectural refinements that improve efficiency and scalability. In particular, the model employs Rotary Position Embeddings (RoPE) (Su et al., 2024) to stabilize training while effectively incorporating positional information into the hidden layers. Furthermore, it integrates Grouped Query Attention (GQA) (Ainslie et al., 2023) to reduce key–value cache size and accelerate inference without sacrificing accuracy. Llama 3 utilizes a pre-layer normalization with RMSNorm (Zhang & Sennrich, 2019), which has been shown to stabilize optimization in large-scale Transformer training. From the above perspective, the architecture of Llama3 is identical to that of other state-of-the-art models, such as Qwen3 (Yang et al., 2025) and Gemma3 (Team et al., 2025). Table 2 describes detailed settings of architecture for 50M (6 layers), 100M (8 layers), and 300M models (12 layers) in our experiments.

Table 2: Settings of model architecture

|  | 50M | 100M | 300M |
|---|---|---|---|
| hidden_size | 192 | 384 | 768 |
| intermediate_size | 768 | 1536 | 3072 |
| num_attention_heads | 3 | 6 | 12 |
| num_hidden_layers | 6 | 8 | 12 |
| num_key_value_heads | 1 | 3 | 4 |
| vocabulary_size | 128,256 | 128,256 | 128,256 |
| training parameters for Vanilla Transformer | 53,086,677 | 119,740,844 | 319,703,886 |
| additional training parameters for our method | 21 | 28 | 78 |

**Pre-training**   We conducted pretraining experiments based on a Llama-3 architecture. The pre-training corpus consisted of a subset of the Fineweb-Edu (Lozhkov et al., 2024) dataset. Regarding the pre-training configuration, sequence length is set to 4096. global batch size is 256, total global training steps are 20,000, corresponding to approximately 20B tokens. We adopt AdamW (Loshchilov & Hutter, 2019) optimizer for the original transformer parameters with a learning rate of 0.0001 without any scheduler (i.e., constant learning rate) and weight decay = 0.01. Regarding vertical attention parameters, we swept over $\{0.0001, 0.001, 0.01, 0.1\}$ for the learning rate and chose 0.01 with weight decay 0.01 after observing the stability and efficacy of vertical attention dynamics. We validated the performance of pre-training based on cross-entropy loss, or perplexity, on both training and validation datasets after splitting the Fineweb-Edu datasets into 99:1.

**Baselines** We compare the performance of our method with the following baseline network architectures, which interact between layers in Transformer: Vanilla Transformer, Skip Layer Attention, Adaptive Layerwise Attention, U-Net Attention, and i-Net Attention. Skip Layer Attention (SLA) (Chen et al., 2024) and Adaptive Layerwise Attention (ALA) (Verma & Pilanci, 2024) dynamically attend to previous layers by modifying the MHA modules so that the query in each layer attend to key and value in the previous layers and calculate the attention weight. Unlike our method, in which the inter-layer attention weights are fixed across samples, these approaches impose constraints on the set of layers to which attention can be applied. U-Net Attention and i-Net Attention adopt a hand-crafted and fixed attention map based on our architecture in order to show how the performance differs from a data-driven attention map as a reference.

- **Vanilla Transformer:** Llama-3 architectur (Dubey et al., 2024) is used as a base architecture. As described in Figure 1, this standard architecture sequentially connects layers. Table 2 describes detailed settings of architecture for 50M, 100M, and 300M models in our experiments.

- **Skip Layer Attention (SLA):** (Chen et al., 2024) introduces a mechanism in transformer architectures in which every layer directly attends to non-adjacent layers, N-distant previous layers, enabling the model to capture dependencies between high-level abstract features and low-level detailed features effectively. More specifically, at each layer, h out of H heads in the MHA module attend to the features of N-distant previous layers, while the rest of (H-h) heads attend to the features of current layers with usual horizontal attention. Based on the default settings in Chen et al. (2024), where the number of inter-layer attention heads is less than or equal to the horizontal heads, and the last few layers attend to the first few layers, we set $N = 4, h = 1$ for 50M model, $N = 6, h = 3$ for 100M model, and $N = 9, h = 6$ for 300M model in our experiments.

- **Adaptive Layerwise Attention (ALA):** (Verma & Pilanci, 2024) introduces a mechanism in transformer-based LLMs where only the final layer dynamically attend to outputs from every K-th previous intermediate layers. Concretely, each intermediate layer produces a hidden representation of the input tokens, and the final layer computes attention weights over these representations, effectively creating a weighted combination of multiple layers' information. The original setting proposed by Verma & Pilanci (2024) requires every head in the final layer to attend to multiple preceding layers. However, this design substantially increases implementation complexity and computational cost. Following the approach of SLA, we modify the architecture such that each head in the final layer attends to a different preceding layer. Due to this modification, we refer to our method as ALA(-). In our expriment, based on the default setting of Verma & Pilanci (2024), we set K=2 across all the size of models.

- **U-Net Attention:** Inspired by the approach proposed in (Videau et al., 2025), in which U-Net architecture (Ronneberger et al., 2015) is adopted into Transformer, we use hand-crafted and fixed attention map implemented on our architecture. Figure 2 (right side) illustrates the example of 300M models.

- **i-Net Attention:** Inspired by the findings in SLA (Chen et al., 2024) that shallow layers interact with deeper layers, we hand-crafted one more attention map in which every layer attends to the first and second layers. Figure 2 (right side) illustrates the example of 300M models. (Videau et al., 2025)

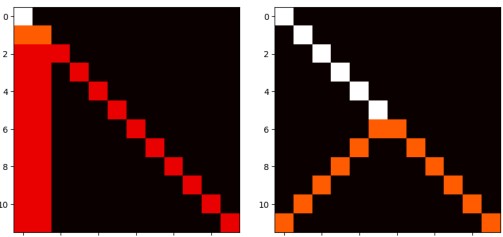

Figure 2: Hand-crafted and fixed attention map used for i-Net Attention (left) and U-Net Attention (right) for 300M models. These two attention maps work on our proposed architecture, i.e., vertical attention.

**Downstream Task Evaluation** We evaluate the pretrained models on a suite of downstream natural language understanding tasks using few-shot in-context learning (ICL) Settings. Specifically, Following prior studies of LLM pre-training (Touvron et al., 2023), we measure performance on seven widely used benchmarks: WINOGRANDE (Sakaguchi et al., 2020), PIQA (Bisk et al., 2020), OPENBOOKQA (Mihaylov et al., 2018), HELLASWAG (Zellers et al., 2019), AI2 ARC-EASY and AI2 ARC-CHALLENGE (Clark et al., 2018), and BOOLQ (Clark et al., 2019). In our experiments, we fix the number of few-shot demonstrations to four across all the benchmarks since the evaluated benchmarks are multiple-choice tasks with either two or four candidate answers. We use the

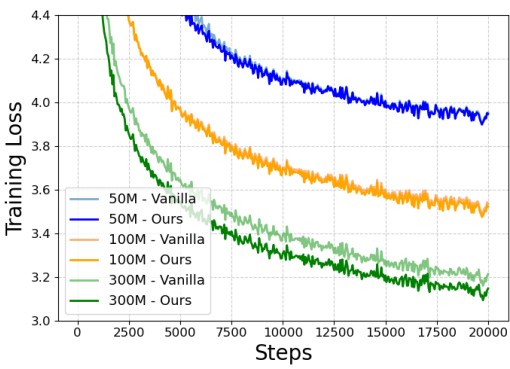

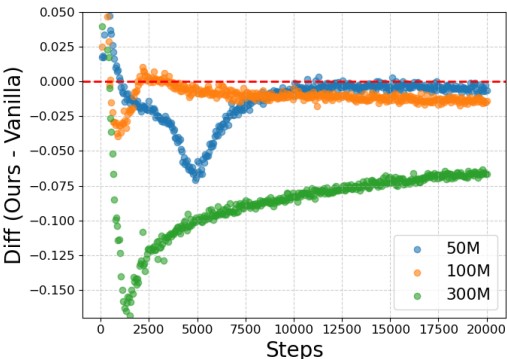

Figure 3: Training loss curve.    Figure 4: Diff. of training loss: Ours - Vanilla.

Table 3: Baseline Comparison. The metric is perplexity.

| Method | 50M | | 100M | | 300M | |
|---|---|---|---|---|---|---|
| | Train | Validation | Train | Validation | Train | Validation |
| Vanilla | 52.09 | 51.00 | 34.35 | 33.80 | 24.87 | 24.40 |
| SLA | 53.25 | 52.02 | 34.52 | 33.95 | 24.64 | 24.20 |
| ALA (-) | 52.93 | 51.76 | 33.93 | 33.40 | 24.61 | 24.16 |
| Ours | 51.77 | 50.73 | 33.87 | 33.30 | **23.26** | **22.89** |
| Ours (U-Attn) | 50.67 | 49.69 | **33.69** | **33.19** | 24.34 | 23.94 |
| Ours (L-Attn) | **50.45** | **49.53** | 33.78 | 33.30 | 25.96 | 25.55 |

`lm-evaluation-harness` framework (Gao et al., 2021) to ensure consistent and standardized evaluation across all tasks.

### 4.2 RESULT

**Pre-training Loss Curve** To demonstrate the stability and consistent superiority of the proposed method during pre-training, Figure 3 presents the training loss curves for the 50M, 100M, and 300M models. While the difference from the Vanilla Transformer is relatively small for the 50M and 100M models, the 300M model exhibits a clear and consistent gap throughout the training steps. To further analyze the stability of the training, Figure 4 illustrates the transition of the loss difference between the Vanilla Transformer and our method, i.e., Ours - Vanilla. In the early stage of training (up to around 5,000 steps), the gap fluctuates unstably, but it eventually converges to a stable value as training progresses.

**Comparison with Baselines** Table 3 shows the result of baseline comparison with our method for the pre-training loss. To clearly understand the difference of each performance, we transformed the metric from cross-entropy loss to perplexity. Although pre-training was conducted for only a single epoch over the training data, we also evaluated perplexity on the validation set, in addition to the training set, to ensure the robustness of the results. Across all model sizes, our method consistently achieves lower perplexity compared to other approaches that employ dynamic inter-layer attention, such as SLA and ALA (-). We also observed that variants of our method with a fixed attention map (U-Net Attention and i-Net Attention) achieve similarly low perplexity. Notably, for the 50M and 100M models, these fixed-attention variants even attain lower perplexity than the versions where the attention map is learned. On the other hand, for the 300M model, learning the attention map results in even lower perplexity. Possible reasons for this include (1) the similarity of the attention maps after training and (2) the sensitivity to the learning rate. These factors are discussed in detail in the following paragraphs.

**Attention Map** Figure 5 illustrates the vertical attention map after pre-training for each model. We observe the following three interesting common patterns across models. First, strong attention

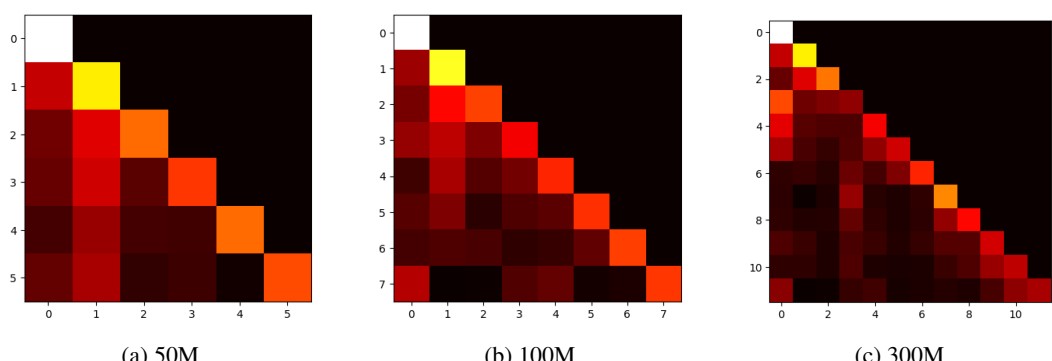

(a) 50M      (b) 100M      (c) 300M

Figure 5: Visualized vertical attention map after pre-training

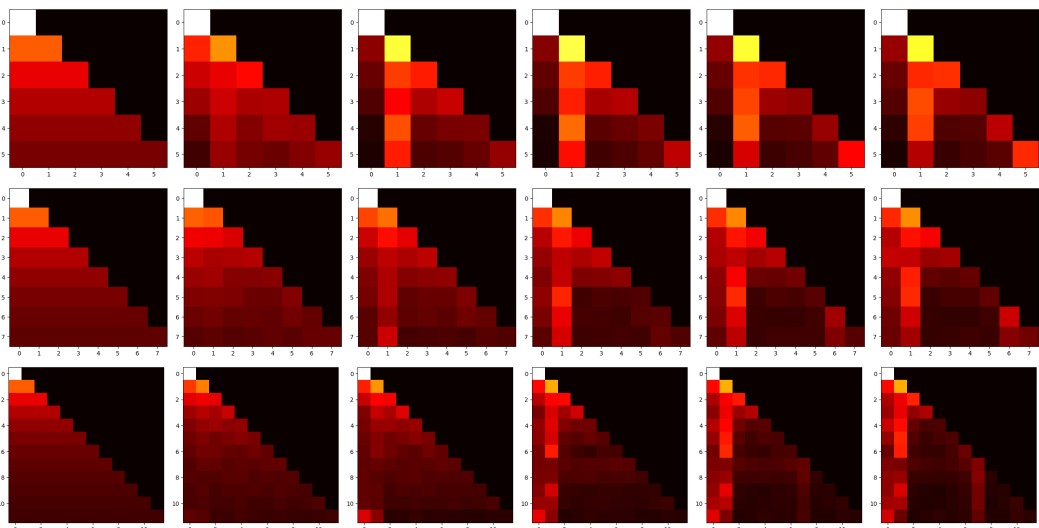

Figure 6: Dynamics of attention map from step0 to step 500 with 100 intervals. Top: 50M, Middle: 100M, Bottom: 300M model. See Appendix Appendix C for the full range of the dynamics.

is applied along the diagonal, indicating that, similar to the Vanilla Transformer, information from recent layers is predominantly incorporated. Second, strong attention is consistently directed toward the lower layers (the first and second layers) across all layers. This suggests that deeper layers recursively integrate information closer to the input during processing. Third, attention to the intermediate layers is relatively weak, exhibiting a sparsity pattern. These observations suggest that the attention map structures of i-Net and U-Net are, to some extent, reasonably organized. However, as the number of layers increases, the optimal attention structure appears to become progressively more complex, giving rise to patterns that cannot be fully captured by hand-crafted designs. **??** illustrates the dynamics of the attention map during pre-training from step 0 to step 500 with 100-step intervals for each model. See Appendix Appendix C for the full range of the dynamics.

**Sensitivity of Learning Rate** Our experiments revealed that the learning rate of the Vertical Attention Module parameters, specifically the parameter used for the logit part of the softmax, has a significant impact on the convergence of the attention maps. Specifically, for the 50M model, we performed a sweep of the learning rate over {0.0001,0.001,0.01,0.1} and trained for 5,000 steps, while monitoring the transition of the attention map entropy, defined as follows:

$$Entropy = \frac{1}{L}\sum_{l \in L}\sum_{i \in L} -\log(\tilde{\alpha}_{l,i})\tilde{\alpha}_{l,i}. \tag{8}$$

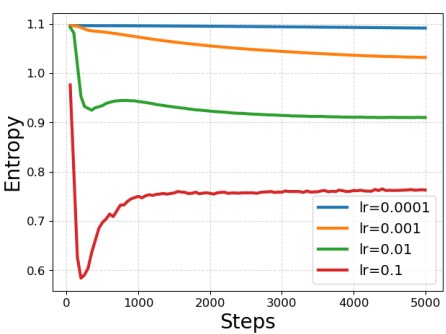

Figure 7: Entropy curve for different learning rates in {0,1, 0.01, 0.001, 0.0001}.

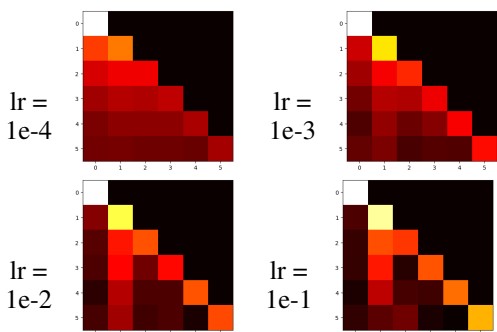

Figure 8: Attention map for different learning rates at 5000 steps.

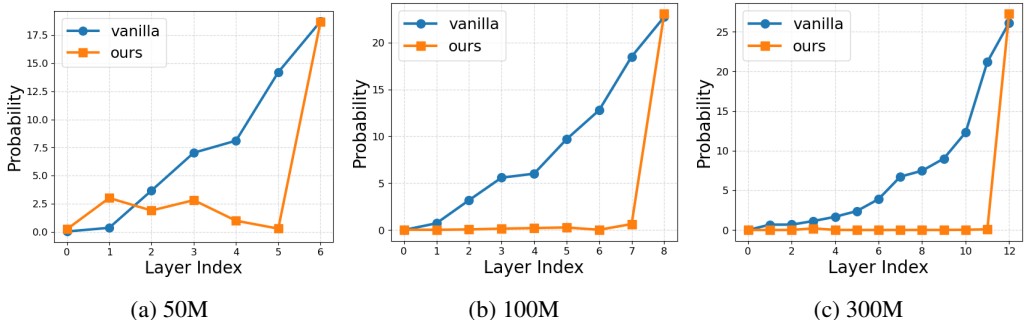

|     (a) 50M     |     (b) 100M    |     (c) 300M    |

Figure 9: Logit lens analysis. We extract output representations from each layer–i.e., input for vertical attention module– and apply layernormalization and linear projection of head parameter.

Intuitively, this formulation indicates the average of entropies across layers in an attention map. The results in Figure 7 showed that with a learning rate of 0.0001 (which is the same as the learning rate used for all other parameters in this experiment), the entropy barely changed, indicating that the attention map was hardly updated. In contrast, a learning rate as high as 0.1 caused large fluctuations in entropy, resulting in unstable learning of the attention map. When the learning rate was set to 0.001 or 0.01, the entropy changed smoothly, suggesting that the attention map was updated appropriately. Figure 8 illustrates attention maps of different learning rates after training 5000 steps, indicating that the attention map convergence greatly differs.

**Logit Lens Analysis**  In order to investigate the internal behavior of models trained with our method, we analyzed the layer-wise representations using the logit lens analysis (Nostalgebraist, 2020; Wendler et al., 2024). Specifically, we extract output representations from each layer–i.e., input representation before vertical attention module– and apply layernormalization and linear projection of head parameter to see the output token probabilities. As shown in Figure 9, in the Vanilla Transformer, the predicted probabilities of output tokens gradually increase as the layers deepen. In contrast, in our Transformer, the predicted probabilities remain almost zero across layers and only sharply increase at the final layer. These results indicate that the internal information-processing behavior of our Transformer is fundamentally different from that of standard, vanilla Transformer. One of the interpretations is that while the vanilla Transformer performs decoding in a roughly linear manner from shallow layers, our Transformer appears to postpone the preparation for decoding by doing encoding process until the final layers through interactions across multiple layers.

**Downstream Task Evaluation**  Table 4 presents a comparison of downstream task performance between the pre-trained Vanilla model and our proposed model, evaluated using few-shot in-context learning (ICL). Across all three model sizes, our proposed model consistently outperforms the Vanilla models on average, indicating the utility and applicability of our method.

Table 4: Downstream task evaluation by few-shot in-context learning setting. Metric is accuracy. HS: HellaSwag, WG: WinoGrande, OBQA: OpenBookQA.

| Model Size | Method | ARC-c | ARC-e | BoolQ | HW | OBQA | PIQA | WG | Avg. |
|---|---|---|---|---|---|---|---|---|---|
| 50M | Vanilla | 17.2 | 34.6 | **48.0** | 26.3 | 13.8 | 57.1 | 51.9 | 35.6 |
| | Ours | **18.5** | 34.5 | 47.2 | 26.3 | **16.4** | 57.0 | **53.8** | **36.2** |
| 100M | Vanilla | **19.0** | **41.8** | 54.3 | 27.0 | 14.4 | 59.0 | 51.0 | 38.1 |
| | Ours | 17.8 | 40.6 | **57.2** | **27.1** | **15.8** | **59.5** | 50.5 | **38.4** |
| 300M | Vanilla | 20.1 | 45.5 | 59.2 | 28.1 | 15.8 | 61.9 | **50.5** | 40.2 |
| | Ours | **20.4** | **47.3** | **59.3** | **28.7** | **17.0** | **62.9** | 49.6 | **40.7** |

## 5 DISCUSSION

Our experiments reveal several important insights regarding inter-layer connectivity in Transformers. First, the Vertical Attention mechanism consistently improves pretraining efficiency and downstream performance, suggesting that learned inter-layer interactions can provide meaningful structural advantages beyond the canonical sequential design. The attention map analysis further indicates that deeper layers selectively integrate information from both the earliest and adjacent layers, which may contribute to more effective feature propagation and representation refinement.

Second, the logit lens analysis highlights a distinct difference in the decoding dynamics between Vanilla and our Transformer. In our model, the preservation of near-zero output probabilities until the final layer suggests that intermediate layers focus on encoding and integrating contextual information rather than producing incremental predictions. This behavior may lead to more concentrated and refined representations in the final layer, potentially explaining the observed improvements in few-shot ICL tasks.

Third, our sensitivity study on the learning rate of Vertical Attention parameters underscores the importance of careful hyperparameter tuning for stable training. Too small a learning rate prevents the attention map from evolving, while too large a rate leads to instability. This observation emphasizes that the inter-layer attention scores play a crucial role in shaping the model's internal dynamics.

Finally, comparing learned attention maps with fixed hand-crafted maps (U-Net and i-Net) shows that data-driven attention provides flexible and scale-adaptive patterns that become increasingly beneficial for larger models. While fixed maps can achieve competitive performance at smaller scales, learning the inter-layer connectivity appears essential for fully leveraging model capacity as the number of layers grows.

## 6 CONCLUSION

We have introduced a novel framework for learning inter-layer connectivity in Transformer models by parameterizing connections as a Vertical Attention mechanism. Our method enables end-to-end optimization of inter-layer paths, allowing the model to dynamically select and weight information from previous layers during training.

Through extensive experiments on LLaMA-style models ranging from 50M to 300M parameters, we demonstrated that our approach consistently achieves lower pretraining loss, improved downstream task performance under few-shot ICL, and generates interpretable attention patterns that reveal interesting structural dynamics. Analyses using logit lens reveal an intriguing prediction pattern–almost zero output prediction until the final layers–, suggesting that our model maintains a more effective encoding of input information until the final layers, supporting robust predictions.

Overall, our work provides both practical performance gains and new insights into the flexible and effective search inter-layer connections of Transformers, highlighting the importance and potential of inter-layer connectivity. Future work includes extending the approach to larger-scale models with billion-sized parameters. Future work also includes applying our method not only to the NLP domain but also to Transformer architectures in other modalities, such as vision and audio. We hope that this study will provide a new perspective for discovering model architectures that significantly outperform and advance beyond the current state-of-the-art Transformers.

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

## A LLM USAGE

We used ChatGPT for support in searching and summarizing related work, drafting initial text, and improving grammar and clarity.

## B ATTENTION MAP

Table 5, Table 6, and Table 7 describe the final attention map of 50M, 100M, and 300M models after training with our method.

## C DYNAMICS OF ATTENTION MAP DURING TRAINING

Figure 10 – Figure 21 illustrate the dynamics of the attention maps during training for 50M, 100M, and 300M models with our method. The tables display the attention maps at every 100 steps, from step 0 to step 19,900.

## D GENERATED TEXT SAMPLES BY MODELS

To evaluate the sample-level quality of the pre-trained models, Table 8, Table 9, and Table 10 present text samples generated by the 50M, 100M, and 300M models pre-trained using our method. We used random sampling with temperature=0.7, top_p=0.9, max_new_tokens= 56 for the setting.

Table 5: Attention map of 50M model

| | | | | | |
|---|---|---|---|---|---|
| 1.00 | 0.00 | 0.00 | 0.00 | 0.00 | 0.00 |
| 0.28 | 0.72 | 0.00 | 0.00 | 0.00 | 0.00 |
| 0.15 | 0.32 | 0.53 | 0.00 | 0.00 | 0.00 |
| 0.14 | 0.30 | 0.12 | 0.45 | 0.00 | 0.00 |
| 0.09 | 0.21 | 0.09 | 0.09 | 0.53 | 0.00 |
| 0.14 | 0.24 | 0.06 | 0.08 | 0.01 | 0.48 |

Table 6: Attention map of 100M model

| | | | | | | | |
|---|---|---|---|---|---|---|---|
| 1.00 | 0.00 | 0.00 | 0.00 | 0.00 | 0.00 | 0.00 | 0.00 |
| 0.22 | 0.78 | 0.00 | 0.00 | 0.00 | 0.00 | 0.00 | 0.00 |
| 0.16 | 0.38 | 0.46 | 0.00 | 0.00 | 0.00 | 0.00 | 0.00 |
| 0.21 | 0.27 | 0.17 | 0.35 | 0.00 | 0.00 | 0.00 | 0.00 |
| 0.08 | 0.24 | 0.11 | 0.16 | 0.42 | 0.00 | 0.00 | 0.00 |
| 0.12 | 0.17 | 0.05 | 0.10 | 0.12 | 0.44 | 0.00 | 0.00 |
| 0.09 | 0.10 | 0.10 | 0.05 | 0.07 | 0.13 | 0.46 | 0.00 |
| 0.25 | 0.00 | 0.01 | 0.11 | 0.14 | 0.02 | 0.03 | 0.45 |

Table 7: Attention map of 300M model

| | | | | | | | | | | | |
|---|---|---|---|---|---|---|---|---|---|---|---|
| 1.00 | 0.00 | 0.00 | 0.00 | 0.00 | 0.00 | 0.00 | 0.00 | 0.00 | 0.00 | 0.00 | 0.00 |
| 0.28 | 0.72 | 0.00 | 0.00 | 0.00 | 0.00 | 0.00 | 0.00 | 0.00 | 0.00 | 0.00 | 0.00 |
| 0.14 | 0.32 | 0.54 | 0.00 | 0.00 | 0.00 | 0.00 | 0.00 | 0.00 | 0.00 | 0.00 | 0.00 |
| 0.48 | 0.15 | 0.17 | 0.20 | 0.00 | 0.00 | 0.00 | 0.00 | 0.00 | 0.00 | 0.00 | 0.00 |
| 0.33 | 0.12 | 0.10 | 0.10 | 0.35 | 0.00 | 0.00 | 0.00 | 0.00 | 0.00 | 0.00 | 0.00 |
| 0.23 | 0.10 | 0.06 | 0.11 | 0.20 | 0.30 | 0.00 | 0.00 | 0.00 | 0.00 | 0.00 | 0.00 |
| 0.06 | 0.07 | 0.05 | 0.15 | 0.09 | 0.18 | 0.42 | 0.00 | 0.00 | 0.00 | 0.00 | 0.00 |
| 0.05 | 0.00 | 0.03 | 0.21 | 0.04 | 0.03 | 0.06 | 0.57 | 0.00 | 0.00 | 0.00 | 0.00 |
| 0.06 | 0.04 | 0.04 | 0.14 | 0.06 | 0.03 | 0.05 | 0.21 | 0.37 | 0.00 | 0.00 | 0.00 |
| 0.10 | 0.07 | 0.03 | 0.10 | 0.07 | 0.04 | 0.07 | 0.11 | 0.11 | 0.30 | 0.00 | 0.00 |
| 0.06 | 0.06 | 0.03 | 0.10 | 0.04 | 0.02 | 0.03 | 0.07 | 0.10 | 0.21 | 0.27 | 0.00 |
| 0.19 | 0.00 | 0.01 | 0.07 | 0.09 | 0.02 | 0.03 | 0.04 | 0.03 | 0.09 | 0.20 | 0.24 |

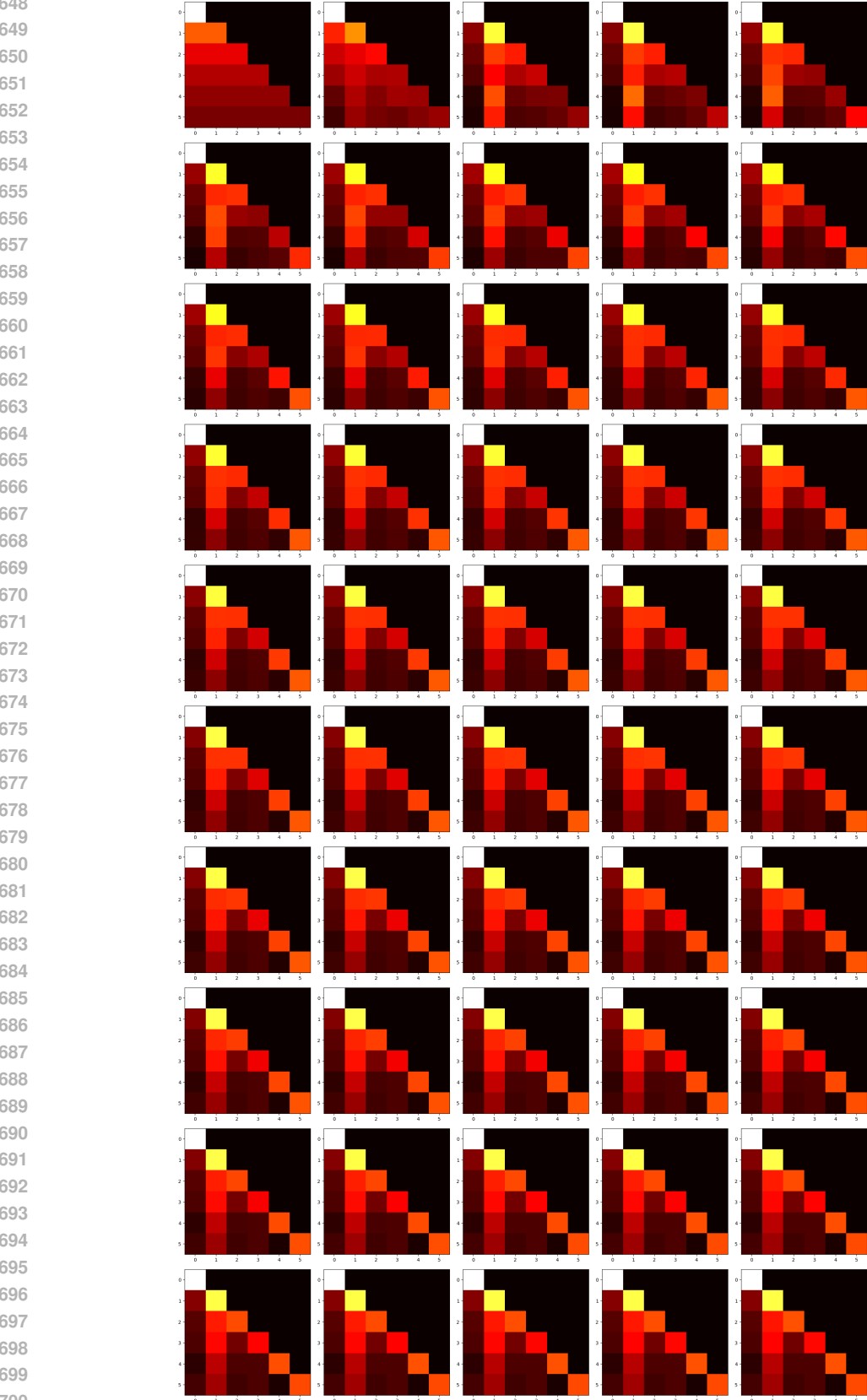

Figure 10: Dynamics of Attention Map - 50M (1/4)

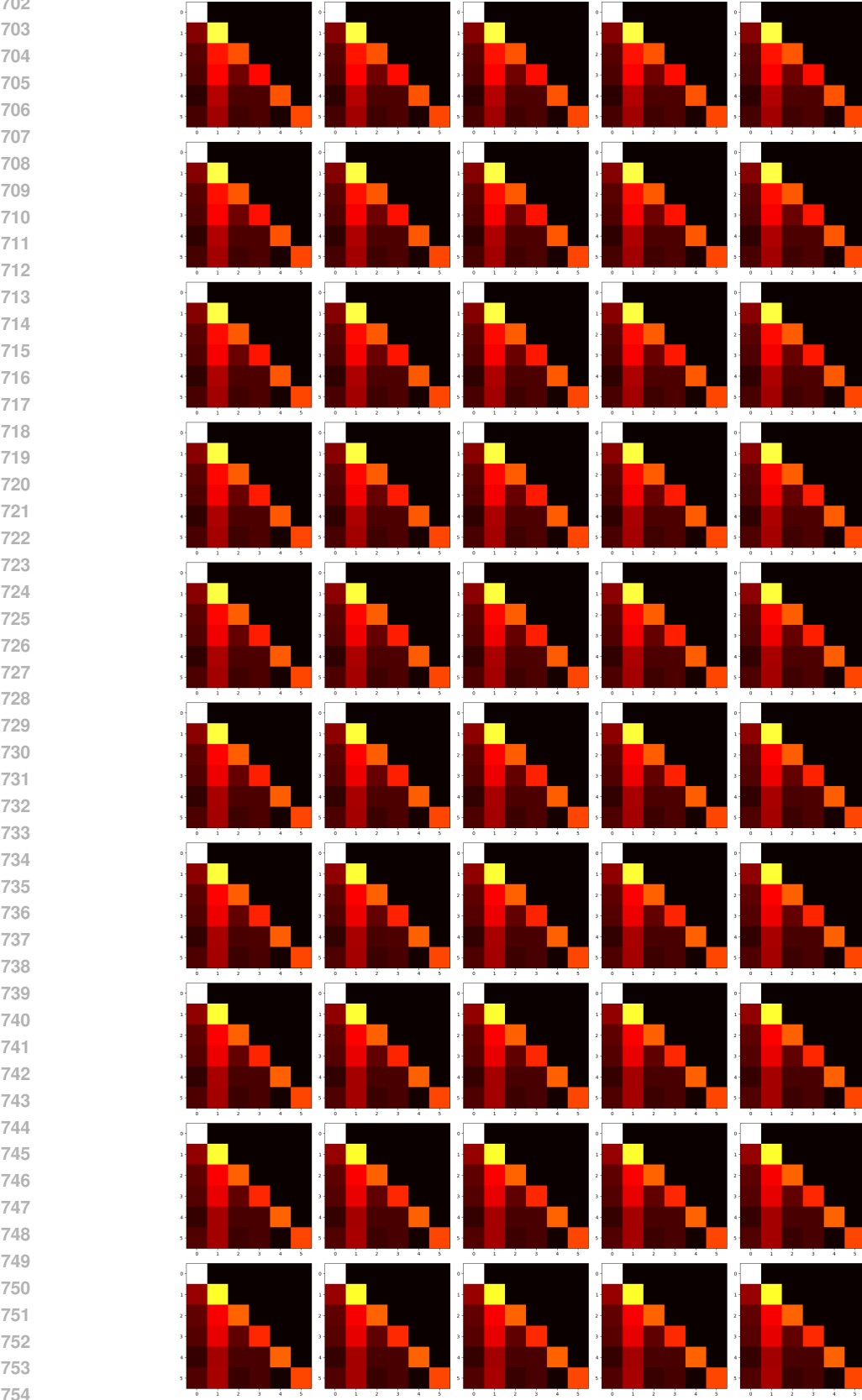

Figure 11: Dynamics of Attention Map - 50M (2/4)

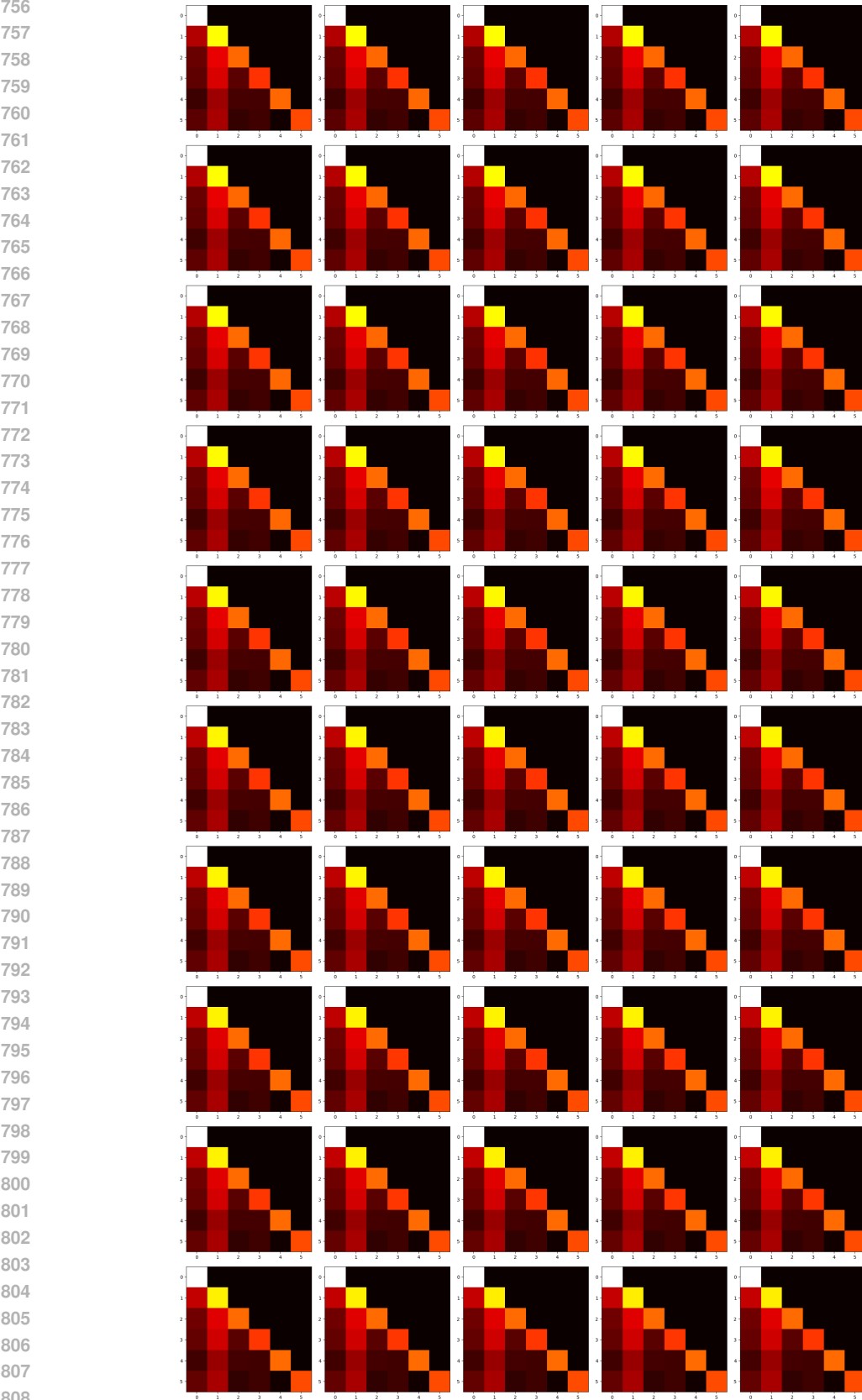

Figure 12: Dynamics of Attention Map - 50M (3/4)

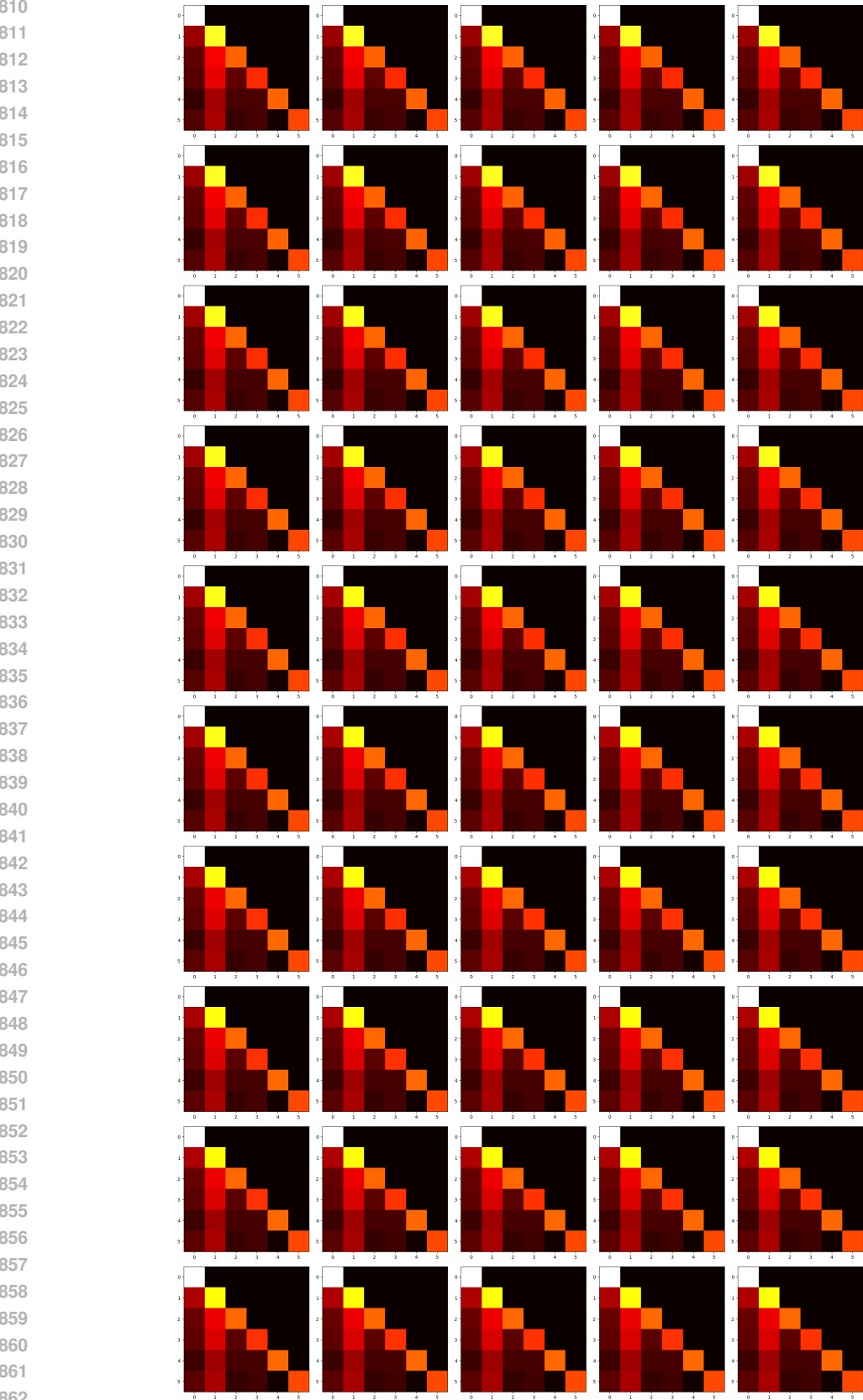

Figure 13: Dynamics of Attention Map - 50M (4/4)

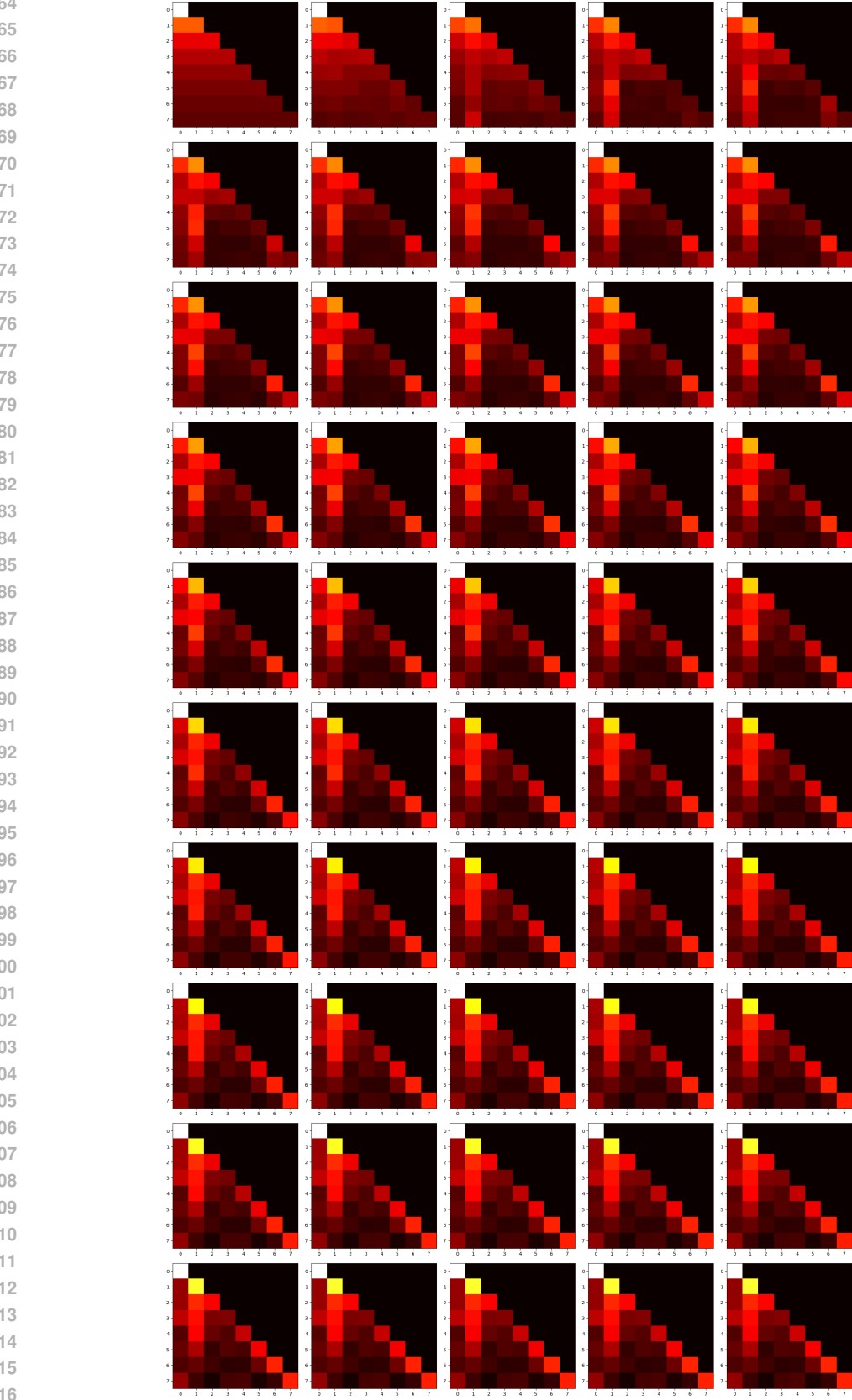

Figure 14: Dynamics of Attention Map - 100M (1/4)

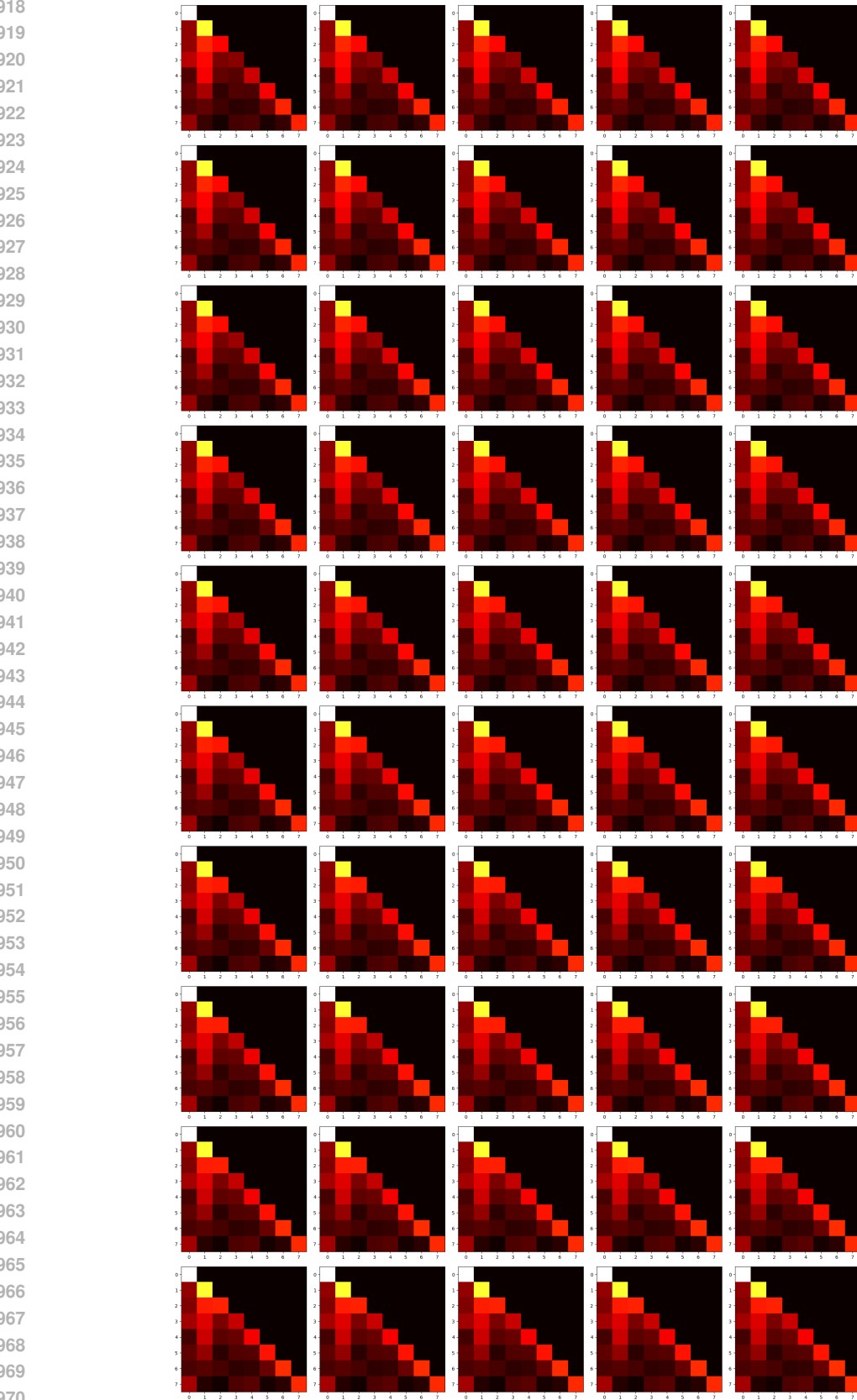

Figure 15: Dynamics of Attention Map - 100M (2/4)

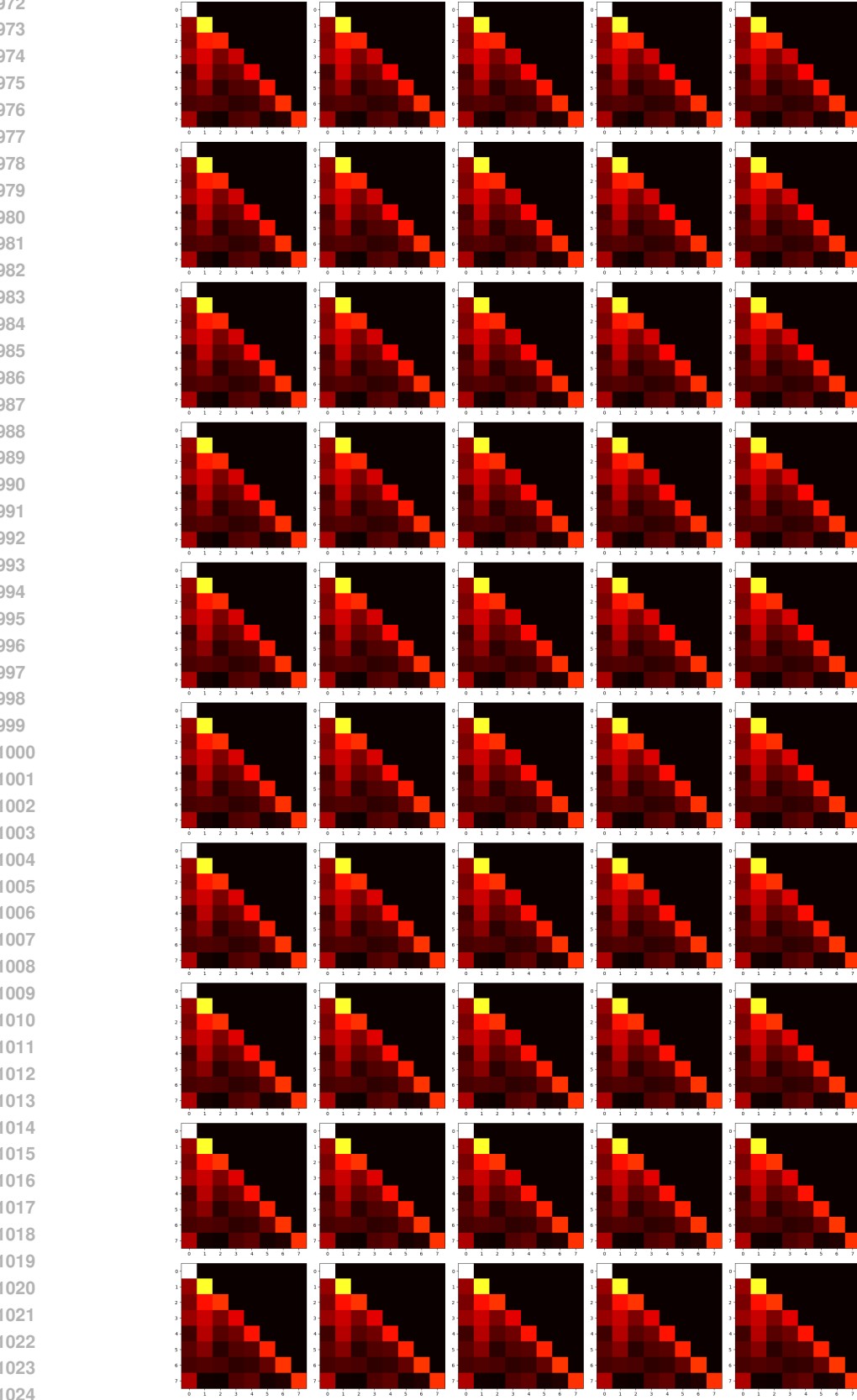

Figure 16: Dynamics of Attention Map - 100M (3/4)

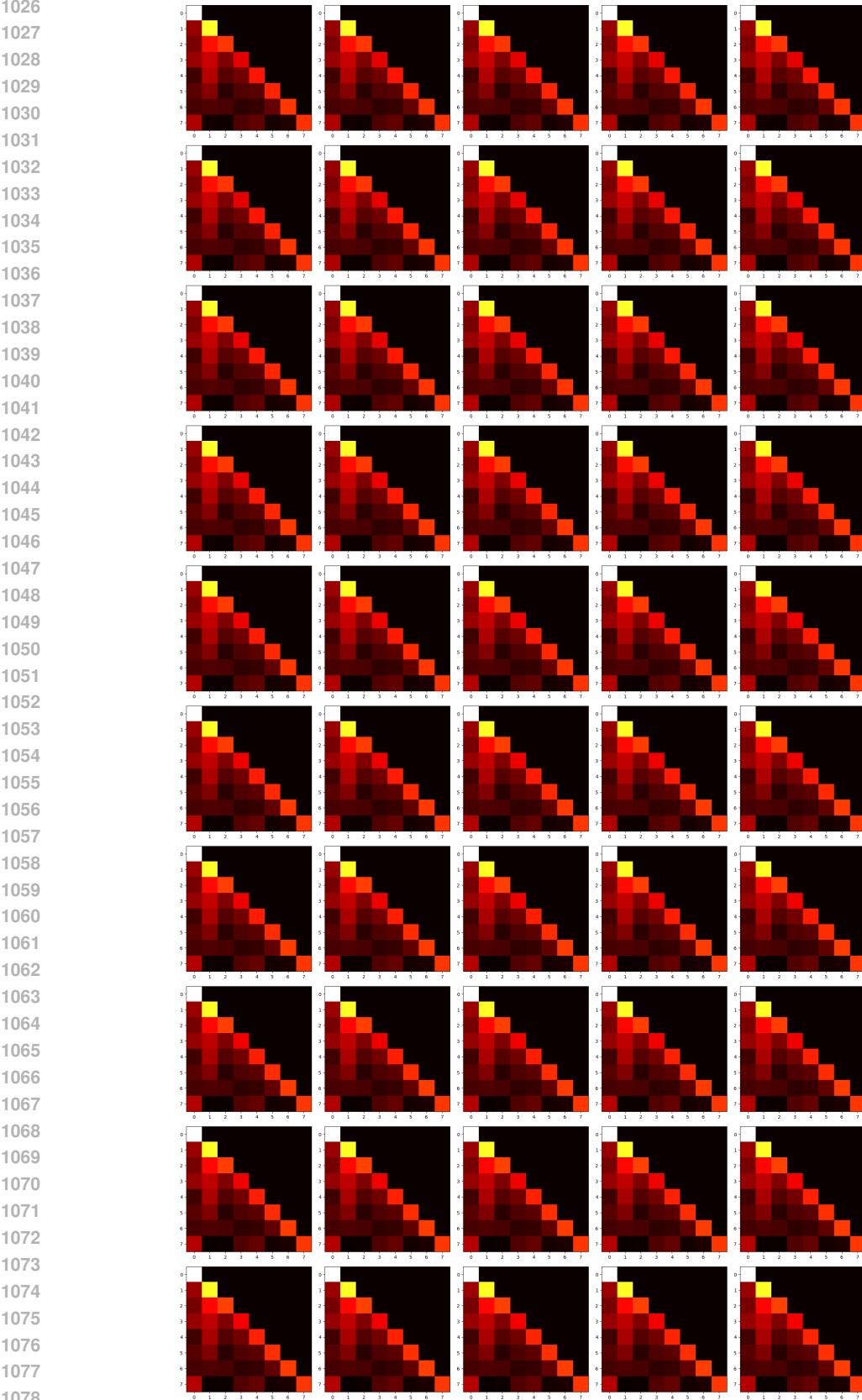

Figure 17: Dynamics of Attention Map - 100M (4/4)

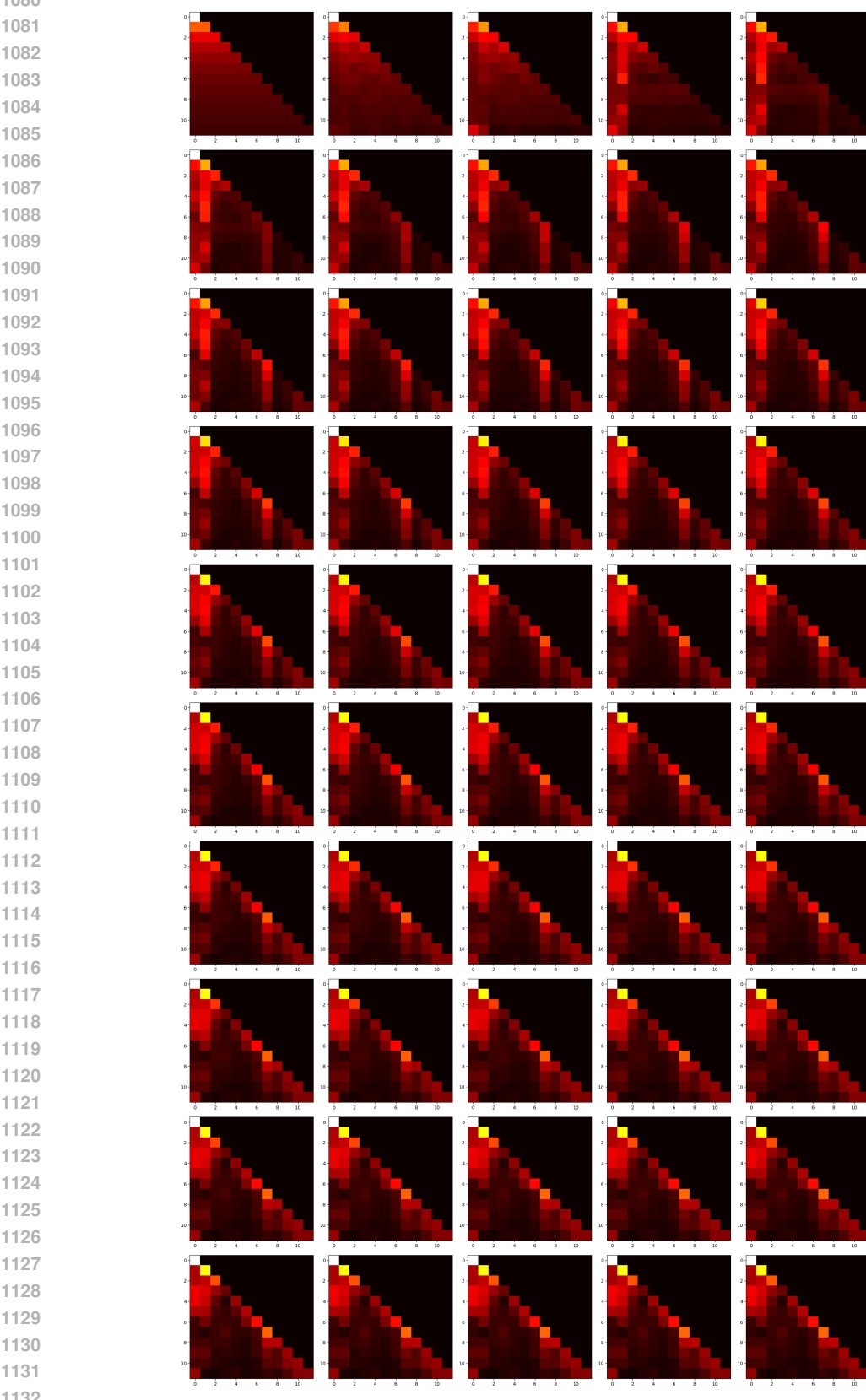

Figure 18: Dynamics of Attention Map - 300M (1/4)

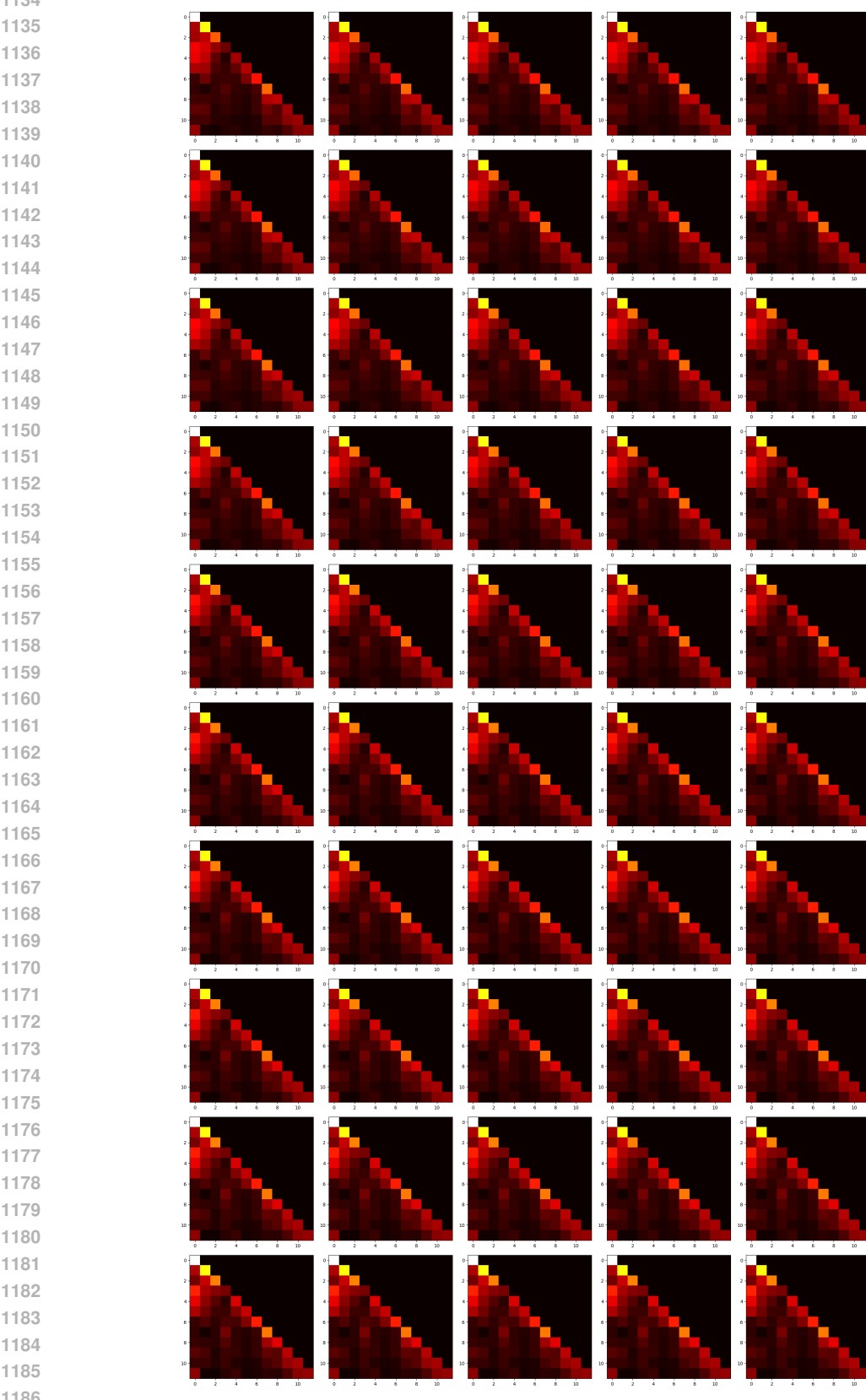

Figure 19: Dynamics of Attention Map - 300M (2/4)

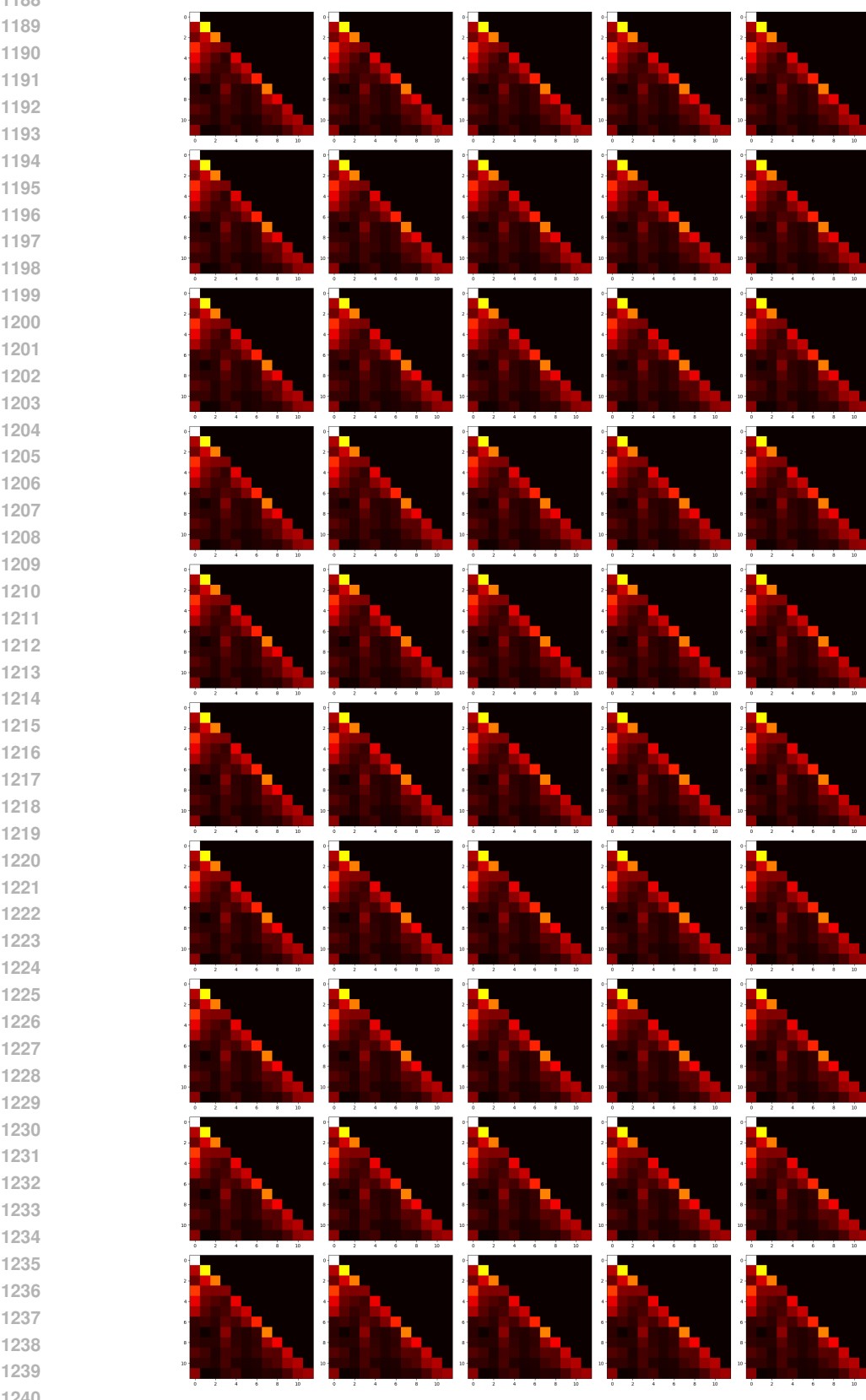

Figure 20: Dynamics of Attention Map - 300M (3/4)

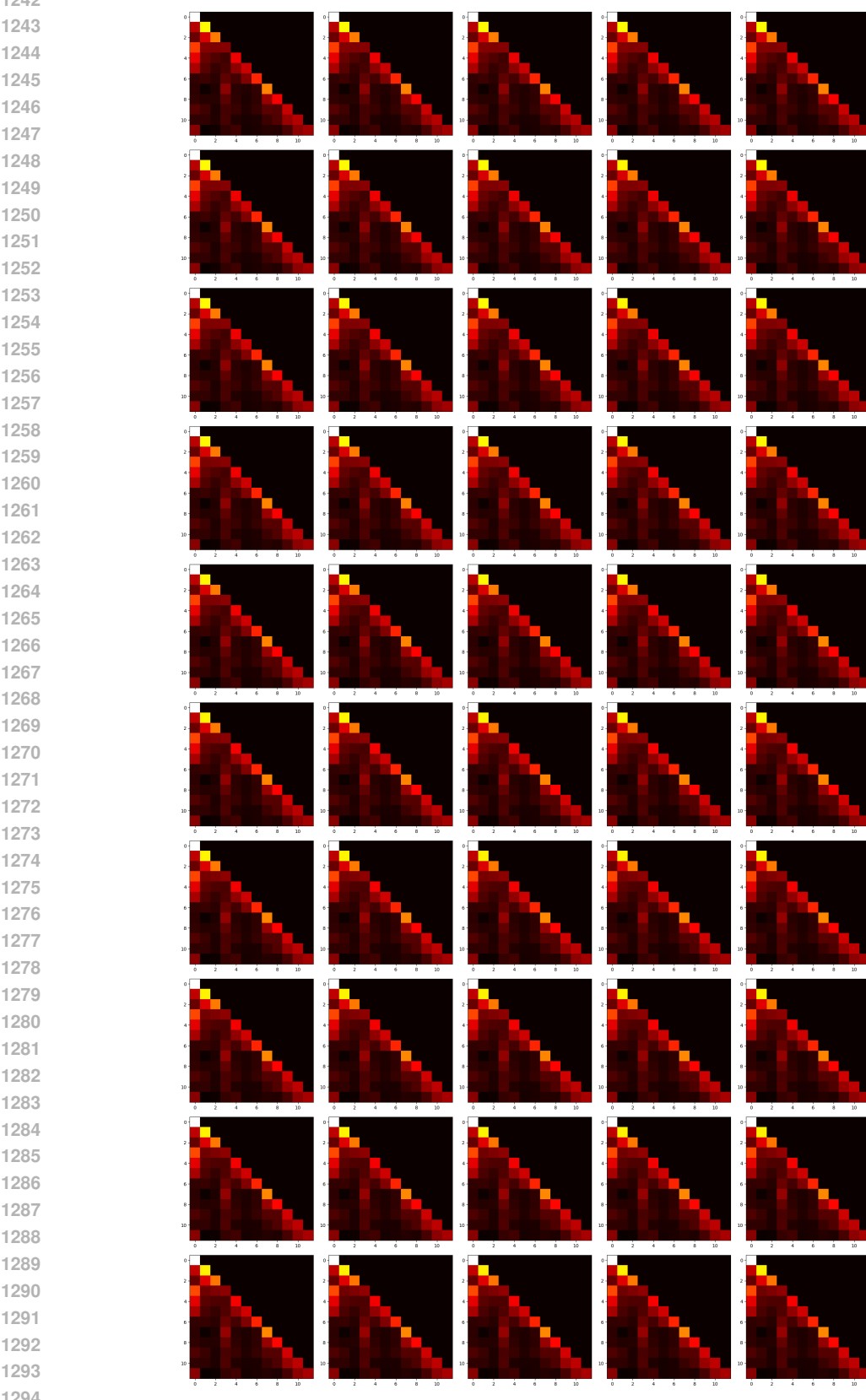

Figure 21: Dynamics of Attention Map - 300M (4/4)

Table 8: Generated text samples from the 50M model. Bold parts indicate the prompts given to the model.

**Hello, world!** All you need to know the answer is "hello" in the plural. As you are speaking, the word is "hello" and we have the answers to the questions, answers to the question, answers to the question, answer to the question, answers to the question, answers to the question, answers to the question, answers to the question, answer to the question, answer to the question, answer to the question, answers to the question, answers to the question, answers to the question, answer to the question, answer to the question, answer to the question, answer to the question, answer to the question, answer to the question, answer to the question, answer to the question, answers to the question, answer to the question, answer to the question, answer to the question, answer to the question, answer to the question, answer to the question, answer to the question, answer to the question, answer to the question, answer to the question, answer to the question, answer to the question, answer to the question, answer to the question, answer to the question, answer to the question, answer to the question, answer to the question, answer to the question, answer to the question, answer to the question, answer to the question, question

**I have a dream.** I have a dream. I have a dream, I have a dream. I have a dream. I have a dream. I have a dream. I have a dream. I have a dream. I have a dream. I have a dream. I have a dream. I have a dream. I have a dream. I have a dream. I have a dream. I have a dream. I have a dream. I have a dream. I have a dream. I have a dream. I have a dream. I have a dream. I have a dream. I have a dream. I have a dream. I have a dream. I have a dream. I have a dream. I have a dream. I have a dream. I have a dream. I have a dream. I have a dream. I have a dream. I have a dream. I have a dream. I have a dream. I have a dream. I have a dream. I have a dream. I have a dream. I have a dream. I have a dream. I have a dream. I have a dream. I have a dream. I have a dream. I

**Once upon a time,** the people of the world are now in a new direction and become aware of their own world. The world is being shaped by a different world, and it's all about the world. In fact, there are two main types of global warming: global warming, global warming, and global warming. The global warming of the world is a global phenomenon that has been studied for many years since the beginning of the 19th century. In the past, the global warming of the world was the beginning of a century. The most recent of the world's major causes was the depletion of the earth. The effects of global warming on the planet were often the result of a combination of global warming, such as climate change, and climate change. These effects were more frequent and more severe. The effects of global warming in the last decades were not even considered to be a major cause of global warming. It is thought that global warming is a serious problem. The world is the world's leading cause of death, the death of people, and the deaths of all over the world. In the past, the world is changing. In 2000, the United States of America, the United States of America, and the United States of America, was a major cause of global warming. The

**New York, Sept. 10 —** The Great Exhibition of 1917 — was a great invention to the British Museum of Art. The exhibition was made up of a series of exhibition displays and posters, and a series of events that inspired a wide variety of artists to learn about the world and its presents. The exhibition was organized by a group of artists and artists, and the exhibition also includes an exhibition of the 1940s and 1940s, the work of the United States. The exhibition was an exhibition of art, and the first exhibition of art to be published in 1921 was created in the 1920s and 1940s. The exhibition included a series of events that the exhibition was designed by the American Museum of Art. The exhibition was organized in 1918, with a total number of exhibitions and exhibitions, including the Museum of Art, the Museum of Art, and the museum. The exhibition is a collection of works that are published in the United States, and includes a series of exhibitions, the exhibition, and a series of exhibitions. The exhibition is a collection of artworks, artworks, and works that were printed in 1917, and is held in the United States. The exhibition was organized in 1930 by a series of exhibitions and exhibitions. The

Table 9: Generated text samples from the 100M model. Bold parts indicate the prompts given to the model.

**Hello, world!** I've been looking at the science of human anatomy, but I'm not sure how to use it in your everyday life. The first part of my post was based on the concept of the world of the human being. The idea of the world of the human being is quite different from the concept of the world of the human being. I'm not sure how this would work, but this is a bit different from the concept of the world of the human being. The idea of the world of the human being is a bit different from the concept of the world of the human being. The concept of the world of the human being is also a bit different from the concept of the world of the human being. It is a different concept because it is a way to express the world in different ways. So what is the difference between the concept of the world of the human being and the concept of the world of the human being? The concept of the world of the human being is a little different from the concept of the world of the human being. The concept of the world of the human being is a little different from the concept of the world of the human being. This concept is very different from the concept of the world of the human being. The concept of the world of

**I have a dream.** I am going to write a book, and I think it is a good idea to do a bit of research. I just want to share this with you. It is a great way to share the ideas and concepts you have been working on. I have read it and told you about the idea, and I want you to share it with you in a way. A lot of people think that you should use a lot of research when you are writing a book, and it is a great way to share your ideas. I have also been doing research, and it is a great way to share your ideas with others. I have a friend who was kind enough to share my thoughts with you, and I have a friend who is very passionate about reading the book. I have also read a book about the author, and I have also read a book about the author, and it is a great way to share ideas with others. I have a friend who is not really interested in writing or writing. I have a friend who is really interested in writing and writing, and it is really a great way to share ideas with others. I have a friend who is interested in writing and writing, and it is really a great way to share ideas with others. A lot of people think

**Once upon a time,** the world would have been filled with many different things. These things were not only the world's original, but also the world's first. The world's first and most important ones were the world's first. This was the world's first, and it was the world's first and most important. It was the world's first. It was the world's first. The world's first. The world's first, it was the world's first. The world's first. The world's first. The world's first. The world's first, it was the world's first, and it was the world's first. The world's first, it was the world's first, it was the world's first, it was the world's first. The world's first, it was the world's first. The world's first. The world's first, it was the world's first, and it was the world's first. The world's first, it was the world's first. The world's first, it was the world's first. The world's first, it was the world's first, it was the world's first. The world's first, it was the world's first, and it was the world's first. The world's first

**New York, Sept. 10 —** The American Red Cross In a 2010 study, a team of researchers from the University of Pennsylvania found that a combination of a diet high in omega-3 fatty acids and a healthy diet can reduce the risk of heart disease. The study, which was published in the journal ACS' Journal of Medicine, found that people who ate a diet high in omega-3 fatty acids were 22 percent more likely to develop atherosclerosis than those who ate a diet low in omega-3 fatty acids. In the study, researchers found that a diet high in omega-3 fatty acids and a healthy diet significantly reduced the risk of developing atherosclerosis. This was because they found a low level of omega-3 fatty acids in the diet. The researchers found that an unhealthy diet also reduced the risk of developing atherosclerosis, but the risk of developing atherosclerosis was reduced by 30 percent. The study is the first to see the benefits of the diets of the 30,000 people who ate the same amount of fish, and the study has yet to be published. The study was published in the journal Circulation. The study was published in the journal Circulation, and the researchers believe the study to be the first to see the benefits of the diets of the

Table 10: Generated text samples from the 300M model. Bold parts indicate the prompts given to the model.

**Hello, world!** I am so busy and hungry and hungry, I am so hungry and hungry, I am so hungry, I am so hungry and hungry, I am so hungry, I am so hungry, I am so hungry and hungry, I am so hungry, I am so hungry, I am so hungry, I am so hungry, I am so hungry, I am so hungry, I am so hungry, I am so hungry, I am so hungry, I am so hungry, I am so hungry, I am so hungry, I am so hungry, I am so hungry, I am so hungry, I am so hungry, I am so hungry, I am so hungry, I am so hungry, I am so hungry, I am so hungry, I am so hungry, I am so hungry, I am so hungry, I am so hungry, I am so hungry, I am so hungry, I am so hungry, I am so hungry, I am so hungry, I am so hungry, I am so hungry, I am so hungry, I am so hungry, I am so hungry, I am so hungry, I am so hungry, I am so hungry, I am so hungry, I am so hungry, I am so hungry, I am so hungry, I am so hungry, I am so hungry, I am so hungry, I

**I have a dream.** I am on the verge of an even brighter future, and that dream is a dream of my life. I am an entrepreneur, and I have a dream of a new generation of workers. I am not sure what the dream will be, but I am sure I will find the answers. I am looking forward to the future of my family and friends, and I hope that I will be able to make a lasting impact on my future. I have a dream of a future where I am a worker. I am the one who will make this dream come true, and I am the one who will create the future for my family. I am the one who will give my life to the world, and I am the one who will make it happen. The dream of my life is a dream of my life. It is a dream of my life that will be realized. It is a dream of my life that will be realized. It is a dream that will be fulfilled by my family and friends, and I am the one who will make this dream come true. The dream of my life is a dream of my life. I am the one who will make this dream come true, and I am the one who will create the future for my family. The dream of my

**Once upon a time,** when people were living in the 20th century, they were walking barefoot in the dirt, the water, and the grass. We have all been here, and we're a part of the 20th century, and so it was, but how did they get to be so? It's not a question of the past. It's a question of the present. This is why we don't have a government that can regulate our actions. That's why we have a government that can control our actions, and what we can do about it. There are two things that we can do about this. The first is to make sure we don't make the same mistakes that we make over and over again. The second is to make sure we don't make the same mistakes that we made in the past. That's why we have a government that can regulate our actions, and what we can do about it.A few years ago, a group of researchers at the University of Toronto decided to try and make a quick, easy way to get a child's attention. In a lab setting, they placed 20 students in a small, plastic tube, and the tubes were placed in a tube that they could see through. They then waited for the students to see

**New York, Sept. 10 —** The world's most famous physicist has been awarded a $1.4 million grant from the National Science Foundation to study how the human body can be manipulated by drugs and other drugs. The $1.4 million grant will help scientists learn more about how the body functions, and how to improve drugs and other drugs to improve people's health. The $1.4 million grant is part of the $1.4 million National Science Foundation's New York-based $1.4 million grant to the National Institute of Neurological Disorders and Stroke to study how drugs and other drugs affect the body's metabolism, immune system and brain. "By studying the metabolic process, we can get a better understanding of how drugs and other drugs affect the body and how these drugs and other drugs affect people's health," said Michael E. C. Martin, a neurologist at the University of California, San Francisco and the lead author of the new research. "By studying the metabolism of drugs and other drugs, we can get a better understanding of how drugs and other drugs affect the body and how these drugs and other drugs affect people's health." The $1.4 million grant will help scientists learn more about how drugs and other drugs affect the body and how they affect the brain, heart and

