# OpenReview forum: "Vertical Attention: Automatic Exploration of Inter-Layer Connections in Transformer-based Language Models"
_ICLR.cc/2026/Conference — Submitted to ICLR 2026_

### Official Review · Reviewer_f6KX · 2025-10-22

**Soundness:** 2
**Presentation:** 2
**Contribution:** 2
**Rating:** 2
**Confidence:** 4

**Summary:**

This paper proposes an architectural variant of the Transformer model that incorporates non-adjacent skip connections. The suggestion is an input-independent attention mechanism used to calculate the weighting coefficients for connecting a layer's output to its preceding layers. These weights are further normalized based on the norm of the respective layer activations, ensuring stability.

Empirical evaluation was performed on a relatively small scale models (up to 300m). The results show an improvement over vanilla Transformer baselines in terms of perplexity and yield (marginal) benefits for in-context learning tasks.

The paper also highlights a difference in the model's internal computation: the probability mass for the next token is observed to arise almost exclusively at the final layer, rather than gradually increasing across successive layers as is typical in the standard Transformer architecture.

**Strengths:**

S1: The core idea of generalizing skip connections beyond adjacent layers is potentially impactful direction for future neural network architectures.

S2: The provided logit lens analysis is intresting. This unusual finding is an important point of discussion, and the authors should elaborate more on the underlying mechanism causing this effect.

S3: The inclusion of non-adjacent connection, such as U-Net and i-Net is important.

**Weaknesses:**

W1: The decision to normalize the skip connection weights by the norm of the layer activations lacks justification. An ablation study is needed to demonstrate the necessity of this specific normalization.

W2: The observed performance differences in few-shot tasks (Table 4) are marginal in my opinion. Furthermore, the evaluation's reliance only on the vanilla Transformer as a baseline, without comparison against other architectures (which the paper considered before), raises questions about the true efficacy of the proposed method.

W3: It is unclear whether the standard adjacent skip connections (found in vanilla Transformers) are retained or omitted when the vertical attention is applied. This must be explicitly clarified, as it is a critical detail of the final architecture.

W4: The paper have some repetitions. Specifically, the discussion of baselines in the related work section overlaps heavily with Section 4, and the content of Section 5 feels redundant with both the introduction (Section 1) and the conclusion (Section 6).

W5 (Minor): There is a missing reference or link reported on line 366.

**Questions:**

See S2 and W3.

---

> ### Author Response · Authors · 2025-11-14
>
> We sincerely appreciate the reviewer's feedback.
>
> > W1: The decision to normalize the skip connection weights by the norm of the layer activations lacks justification. An ablation study is needed to demonstrate the necessity of this specific normalization.
>
> We agree that we need an ablation study to demonstrate the necessity of this normalization.
> We will conduct such additional experiments for the revised manuscript in the future.
>
> > W2: The observed performance differences in few-shot tasks (Table 4) are marginal in my opinion. Furthermore, the evaluation's reliance only on the vanilla Transformer as a baseline, without comparison against other architectures (which the paper considered before), raises questions about the true efficacy of the proposed method.
>
> As the reviewer points out, some of the benchmark results are nearly chance rate, so we agree that we need more training to make a clear difference. In addition we agree that we need more baseline experiments also for the downstream tasks.
>
> > W3: It is unclear whether the standard adjacent skip connections (found in vanilla Transformers) are retained or omitted when the vertical attention is applied. This must be explicitly clarified, as it is a critical detail of the final architecture.
>
> When the vertical attention is applied, adjacent skip connections strongly exist. See the diagonal elements of the attention map in Figure 5.
>
> > W4: The paper have some repetitions. Specifically, the discussion of baselines in the related work section overlaps heavily with Section 4, and the content of Section 5 feels redundant with both the introduction (Section 1) and the conclusion (Section 6).
>
> Thank you for the suggestion. In future revisions of this paper, we will eliminate such redundancy and add more important information.
>
> > W5 (Minor): There is a missing reference or link reported on line 366.
> Thank you for the notification. It is Figure 6.
>
> Again, thank you so much for the valuable feedback. We will incorporate it in the future revised paper.

---

> > ### Comment · Reviewer_f6KX · 2025-11-25
> >
> > Thanks for your response. Good Luck with the re-submission.

---

### Official Review · Reviewer_GHvq · 2025-10-31

**Soundness:** 3
**Presentation:** 1
**Contribution:** 1
**Rating:** 2
**Confidence:** 5

**Summary:**

This paper proposes to enhance the standard transformer architecture with "vertical attention" -- inter-layer weight scalars that can flexibly adjust the connectivity between layers. The paper then evaluates this architectural modification on three small-scale generative language models and closely looks into the learned connectivity.

**Strengths:**

The proposed method of learning inter-layer weighting is an interesting extension of the standard sequential transformer architecture. It could lead to a more effective layer connectivity, or it can simply increase the expressiveness of transformer while adding a negligible number of additional parameters.

The authors test this method on three small generative language models that are evaluated on standard benchmarks via 4-shot in-context learning. Although calling this "large-scale experiments" (in the abstract) is a clear overstatement.

The paper very detailedly shows the dynamics of the learned weights throughout training, which nicely illustrates the proposed method. Unfortunately, it does not really analyze or discuss what the learned connectivity implies for the future transformer architectures.

**Weaknesses:**

The paper copies previous works without referencing them, In particular, the "vertical attention" formulation is exactly the same as the learned layer-selection mechanism in *"Not all layers are equally as important: Every Layer Counts BERT (2023)"* [1] and closely related to the one proposed in *"DenseFormer: Enhancing Information Flow in Transformers via Depth Weighted Averaging (2024)"* [2].

Learning layer-weighting scalars and normalizing them (Equations 1 and 2) is exactly the same as the main method proposed in [1; Equation 6 in that paper), normalization of the hidden vectors is also proposed in [1] as one of the tested variants (Section 5.2, variant 2 in that paper). DenseFormer [2] proposes a very similar weighting mechanism, they only omit the softmax normalization of the weight scalars (Section 3 in that paper). Both papers also visualize the learned weighting patterns in the same way as this paper, by showing the triangular heatmaps (Figure 1 in [1] and Figure 4 in [2]).

In summary, while the plagiarization might be unintended, there is still no real contribution of this work compared to the previous papers published on this topic. In particular, the paper [2] studies this mechanism on a larger scale and also proposes ways to reduce its computational overhead, a problem not even mentioned in this paper.

_____

**Other issues:**
- Doesn't the baseline comparison in Table 3 show that your "vertical attention" is not very useful? Instead of learning the weights automatically, it is possible to fix them as proposed by earlier work (U-Net and i-Net) and achieve better performance (also with lower computational overhead) as demonstrated in 2 out of 3 training runs.
- All evaluated systems are worse or equal to the random baseline on both ARC-challenge (~25%), OpenBookQA (~25%) and WinoGrande (~50%). What is the point of comparing the method on such benchmarks? It also seems that the models might be severely undertrained when they are not capable of outperforming random guessing in 4-shot ICL evaluation.
- It is not true that the transformer architecture of Llama3 (used in this paper) is identical to Qwen3 and Gemma3 (line ~193), there are many differences in fact.
- There are quite a few typos, it would be good to properly check the writing.
_____

**References:**
- [1] Not all layers are equally as important: Every Layer Counts BERT -- https://aclanthology.org/2023.conll-babylm.20/
- [2] DenseFormer: Enhancing Information Flow in Transformers via Depth Weighted Averaging -- https://proceedings.neurips.cc/paper_files/paper/2024/hash/f67449c7ab72f441d3a713b046c6818c-Abstract-Conference.html

**Questions:**

- You propose learning layer weights as a means of doing "network architecture search", have you thus tried to turn the learned patterns into fixed connections? For example, judging from Figure 5, it might make sense to take a regular transformer and only add connections to the first and fourth layers. Is that maybe something you have experimented with?
- How do you initialize the layer weights?

---

> ### Author Response · Authors · 2025-11-14
>
> We sincerely appreciate the reviewer's feedback.
>
> > The paper copies previous works without referencing them, In particular, the "vertical attention" formulation is exactly the same as the learned layer-selection mechanism in "Not all layers are equally as important: Every Layer Counts BERT (2023)" [1] and closely related to the one proposed in "DenseFormer: Enhancing Information Flow in Transformers via Depth Weighted Averaging (2024)" [2].
>
> We have checked the above papers and confirmed that our proposal and experiment design / results are quite similar despite the difference regarding in-depth analysis and model architecture. We agree that our paper needs a substantial rewrite to pivot the contributions of the paper. We deeply apologize for the insufficient investigation of prior research.
>
> > Doesn't the baseline comparison in Table 3 show that your "vertical attention" is not very useful? Instead of learning the weights automatically, it is possible to fix them as proposed by earlier work (U-Net and i-Net) and achieve better performance (also with lower computational overhead) as demonstrated in 2 out of 3 training runs.
>
> Certainly, our experiments demonstrate that in smaller models (50M, 100M; models with small number of layers), fixed attentions like U-Net and i-Net show better performance. However, as the models become large (300M; models with large number of layers), dynamic learnable attention achieves the better performance. As the reviewer points out, we need more experiments to verify the reliability of this trend.
>
> > All evaluated systems are worse or equal to the random baseline on both ARC-challenge (25%), OpenBookQA (25%) and WinoGrande (~50%). What is the point of comparing the method on such benchmarks? It also seems that the models might be severely undertrained when they are not capable of outperforming random guessing in 4-shot ICL evaluation.
>
> We follow the convention of using this set of benchmarks for pre-training research (e.g. https://arxiv.org/abs/2507.12466, https://arxiv.org/abs/2503.00808, https://arxiv.org/abs/2404.01204).
> However, as the reviewer points out, some of the benchmark results are nearly chance rate, so we agree that we need more training to make a clear difference.
>
> > It is not true that the transformer architecture of Llama3 (used in this paper) is identical to Qwen3 and Gemma3 (line ~193), there are many differences in fact.
>
> We apologize for the misleading description in the paper. We agree that Llama3 is not identical to Qwen3 or Gemma3.
> These models just share several important common features, such as RoPE, GQA, RMSNorm, etc.
>
> > You propose learning layer weights as a means of doing "network architecture search", have you thus tried to turn the learned patterns into fixed connections? For example, judging from Figure 5, it might make sense to take a regular transformer and only add connections to the first and fourth layers. Is that maybe something you have experimented with?
>
> Thank you for the suggestion. We will conduct such additional experiments for the revised manuscript in the future.
>
> > How do you initialize the layer weights?
>
> We initialize the layer weights (s in equation(1)) with zero so that the attention weight becomes uniform at the beginning of training.
>
> Again, thank you so much for the valuable feedback. We will incorporate it in the future revised paper.

---

### Official Review · Reviewer_XTaY · 2025-11-01

**Soundness:** 3
**Presentation:** 3
**Contribution:** 1
**Rating:** 2
**Confidence:** 5

**Summary:**

The authors explore how letting the model choose which previous layer is important for the current layer changes performance and the internal behaviour of the model. To do this, they add a few parameters that change the input of a layer to the normalised weighted sum of all the previous layers. They show that by doing this, the model performs better both during pretraining and on downstream tasks. They also investigate how these weights evolve during training. Finally, they show that the internal behaviour of the model changes as the calculations of the output logits are left to the final layer.

**Strengths:**

- The paper is well-written and easy to follow.
- Multiple sizes of models are tested, and the architecture modification performance improvement not only holds but also increases as models get larger.
- The authors do an in-depth analysis of the evolution of the layer weights during training and show that common patterns such as early layer importance emerge at all sizes, but that as the model size increases, the patterns become more complicated.
- The authors compare to various baselines and show that using learned model weights outperforms fixed weights.
- The authors show that the internal behaviour of the models changes with the addition of layer weighting, helping to potentially explain the success in using layer weighting.
- The authors show that the improvement in performance holds for downstream tasks.

**Weaknesses:**

The biggest problem of this paper is that its architecture is not novel; it has been done by both ELC-BERT (Georges Gabriel Charpentier & Samuel, CoNLL-BabyLM 2023) and DenseFormer (Pagliardini, NeurIPS 2024), with the method in this paper being the same as strict normalisation in ELC-BERT. While the paper does a more in-depth analysis of the architecture and applies it to a decoder rather than an encoder, or encoder-decoder, it would require a substantial rewrite to pivot the contributions of the paper.

[Not all layers are equally as important: Every Layer Counts BERT](https://aclanthology.org/2023.conll-babylm.20/) (Georges Gabriel Charpentier & Samuel, CoNLL-BabyLM 2023)

Matteo Pagliardini, Amirkeivan Mohtashami, Francois Fleuret, and Martin Jaggi. 2025. DenseFormer: enhancing information flow in transformers via depth weighted averaging. In Proceedings of the 38th International Conference on Neural Information Processing Systems (NIPS '24), Vol. 37. Curran Associates Inc., Red Hook, NY, USA, Article 4336, 136479–136508.

**Questions:**

- When adding previous layers, do you add the output of the layer before the residual connection from the input to the block or after (this actually affects the importance of each layer, since in a vanilla transformer, you can see the input to each layer being an equal sum of all the previous layers (when post-norm is applied) due to the residual)?
- Line 366: Should be Figure 6 and not ??
- Did you observe longer training times for the models (as found in other papers)?

---

> ### Author Response · Authors · 2025-11-14
>
> We sincerely appreciate the reviewer's feedback.
>
> > The biggest problem of this paper is that its architecture is not novel; it has been done by both ELC-BERT (Georges Gabriel Charpentier & Samuel, CoNLL-BabyLM 2023) and DenseFormer (Pagliardini, NeurIPS 2024), with the method in this paper being the same as strict normalisation in ELC-BERT. While the paper does a more in-depth analysis of the architecture and applies it to a decoder rather than an encoder, or encoder-decoder, it would require a substantial rewrite to pivot the contributions of the paper.
>
> We have checked the above papers and confirmed that our proposal and experiment design / results are quite similar despite the difference regarding in-depth analysis and model architecture. We agree that our paper needs a substantial rewrite to pivot the contributions of the paper. We deeply apologize for the insufficient investigation of prior research.
>
> > When adding previous layers, do you add the output of the layer before the residual connection from the input to the block or after (this actually affects the importance of each layer, since in a vanilla transformer, you can see the input to each layer being an equal sum of all the previous layers (when post-norm is applied) due to the residual)?
>
> We add each previous layer's output "after" the residual connection. Certainly, we have two choices (before or after). We will consider one more time regarding the choice.
>
> > Line 366: Should be Figure 6 and not ??
>
> Yes, thank you for the notification.
>
> > Did you observe longer training times for the models (as found in other papers)?
>
> Yes, actually if we have deeper layers it takes longer training times even though the total FLOPs are marginal increase.
> There are non-negligible overhead time for creating new variables on memory of GPU as it increases with O(L^2).
>
> Our experiment revealed that the vertical attention weights for the lower layers (1st and 2nd layers) and the immediately preceding layers are relatively high. Therefore, we are currently thinking that by leveraging this inductive bias to sparsify the attention weights from the beginning of training and limit the trainable attention parts, we can shorten training time while maintaining the performance.
>
> Again, thank you so much for the valuable feedback. We will incorporate it in the future revised paper.

---

### Author Response · Authors · 2025-11-14
**To all the reviewers**

We sincerely appreciate all the reviewers' feedback.

We have checked the papers raised by reviewer XTaY and GHvq, confirming that our proposal and experiment design / results are quite similar despite the difference regarding in-depth analysis and model architecture. We think that our paper needs a substantial rewrite to pivot the contributions of the paper. We deeply apologize for the insufficient investigation of prior research.
- [1] Not all layers are equally as important: Every Layer Counts BERT -- https://aclanthology.org/2023.conll-babylm.20/
- [2] DenseFormer: Enhancing Information Flow in Transformers via Depth Weighted Averaging -- https://proceedings.neurips.cc/paper_files/paper/2024/hash/f67449c7ab72f441d3a713b046c6818c-Abstract-Conference.html

Once again, we sincerely thank all the reviewers for taking the time to review our work and for the valuable feedback.
We will incorporate it in the future revised paper.

---

### Meta-Review · Area_Chair_tpCP · 2026-01-02

**Summary:**

Reviewers XTaY and GHvq both identified that the authors' proposed architecture and experiment design are very similar to that of prior published work.  The authors concurred with this assessment, and agreed that the paper needs a serious revision - for this reason, I recommend rejection.

**Reviewer Concerns:**

The reviewers XTaY and GHvq both identified that prior work had already investigated a nearly identical architectural choice, and this cannot be addressed during a rebuttal.

**Reviewer Scores:**

I think that each reviewer would have likely maintained or lowered their score of 2, given the existence of prior work that was quite similar (as confirmed by the authors).

---

### Decision · Program_Chairs · 2026-01-26

Reject